# Nitroreductase-triggered indazole formation

Henrik Terholsen ⬡, Lisa Medema, Elizaveta Chernyshova ⬡,
Alejandro Prats Luján, Gerrit J. Poelarends ⬡ & Sandy Schmidt ⬡ ✉

Biocatalysis contributes significantly to the development of more sustainable synthetic pathways by using mild reaction conditions and water as a solvent. However, many relevant classes of compounds, including privileged groups in drug design, are not yet accessible via enzymatic pathways. In this context, the development of an enzymatic route to indazoles remains an unmet challenge. Here, we present a nitroreductase-triggered indazole formation, in which 2-nitrobenzylamine derivatives are converted to reactive nitrosobenzylamine intermediates that spontaneously cyclize and aromatize to indazoles. Two nitroreductases accept a series of 2-nitrobenzylamine derivatives with excellent conversions (up to >99 %). In the case of N-substituted nitrosobenzylamines, 2H-indazoles are formed, whereas other derivatives led to 1H-indazoles. The synthetic value of the nitroreductase-triggered indazole formation is further demonstrated by successful coupling with an imine reductase in a sequential cascade reaction on a 50 mg scale. With this cascade, 2H-indazoles are accessible from cheap 2-nitrobenzaldehyde and primary amines, resulting in up to 85 % conversion and 68 % isolated yield.

It is a well-known phenomenon in drug discovery that many bioactive compounds share the same scaffold[1,2]. Among these "privileged groups", bicyclic N-heterocycles such as benzimidazoles, benzoxazoles, and (tetrahydro)isoquinolines are overrepresented, making them very important targets for synthetic strategies. Consequently, many of these privileged groups have already been accessed via biocatalytic routes, which are often discussed in the context of sustainable synthesis[3–6]. However, enzymatic routes have not been described for all N-heterocyclic groups, such as in the case of indazoles. These molecules are found in many commercially successful drugs, such as granisetron, axitinib, niraparib and pazopanib, which are used to treat nausea and cancer (Fig. 1A). The 1H- and 2H-tautomers of indazoles are equally important for drug development. In the case of the N-unsubstituted indazoles, the 1H-tautomer dominates because it is more stable[7–9].

Due to the importance of both indazole forms, several synthetic routes have been developed for the preparation of 1H- and 2H-indazoles (Fig. 1B)[9]. The most commonly used synthetic strategies for the synthesis of 1H-indazoles include acylation, nitrosation, and cyclization of o-toluidine[10] and intramolecular amination of 1-aryl-2-(2-nitrobenzylidene)hydrazines[11]. The Cu-catalyzed sequential condensation of 2-bromobenzaldehyde, a primary amine and azide[12]

and the palladium-catalyzed cyclization of N-aryl-N-(o-bromobenzyl) hydrazines are used to form 2H-indazoles[13]. Furthermore, organophosphorus-catalyzed pathways for forming 2H-indazoles were developed starting from 2-nitrobenzaldehydes and primary amines[14,15]. 3-Alkoxy-group-functionalized 2H-indazoles can be generated via the classical, base-catalyzed Davis-Beirut reaction. This reaction relies on an intramolecular, redox reaction of 2-nitrobenzylamines, followed by the addition of an alcohol at the 3-position of the indazole ring[16]. A photochemical acid-catalyzed variation has been established that make these products accessible from 2-nitrobenzyl alcohols, primary amines, and alcohols[17]. Additionally, a Lewis acid-catalyzed approach for the synthesis of 2H-indazoles from imines of 1-(2-aminophenyl)ethenone derivatives has been published recently[18].

These traditional chemical and electrochemical methods to prepare indazoles suffer from drawbacks such as the need for hazardous hydrazine derivatives, the use of noble metal catalysts or redox cells, and the employment of often harsh reaction conditions (Fig. S1).

Enzyme catalysis can be an attractive alternative for the synthesis of 1H- and 2H-indazoles, especially to provide a synthetic route with a lower environmental impact. At the same time, it can facilitate the expansion of the indazole synthesis toolbox and enable enzymatic

Department of Chemical and Pharmaceutical Biology, Groningen Research Institute of Pharmacy, University of Groningen, Groningen, The Netherlands.
✉e-mail: s.schmidt@rug.nl

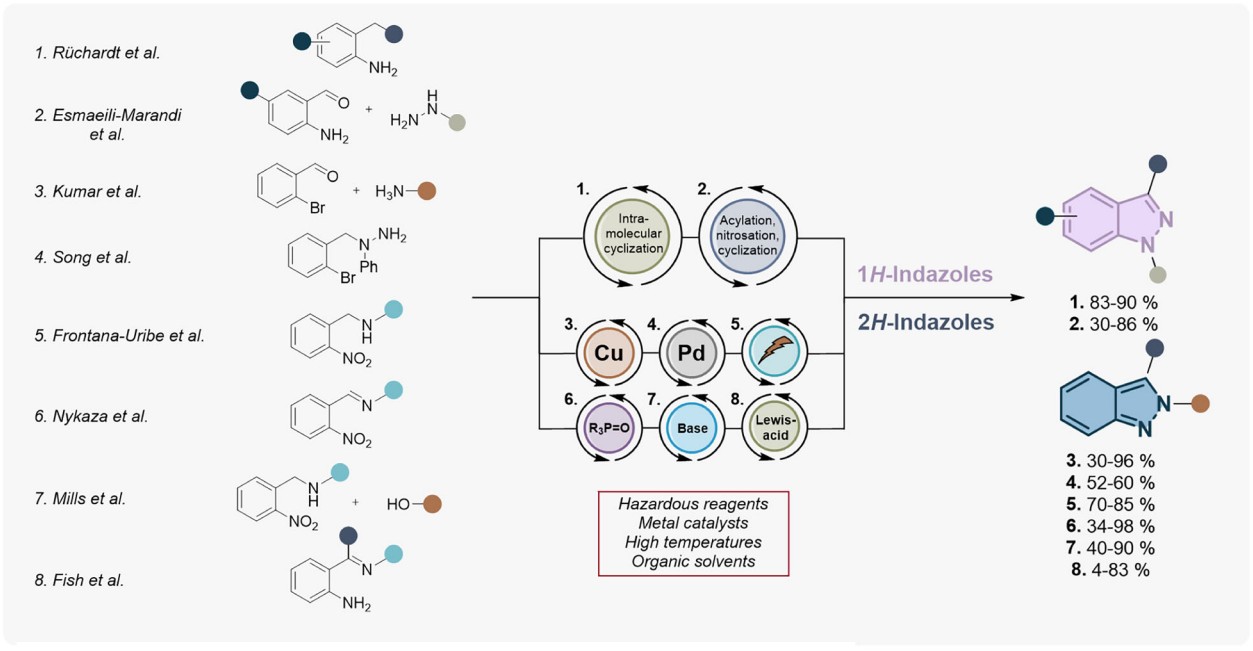

A | 1H- and 2H-indazoles in common pharmaceuticals

**Granisetron** (anti-nausea)

**Axitinib** (anti-cancer)

**Pazopanib** (anti-cancer)

**Niraparib** (anti-cancer)

B | Previous work: Chemical strategies to synthesize 1H- and 2H-indazoles

1. Rüchardt et al.
2. Esmaeili-Marandi et al.
3. Kumar et al.
4. Song et al.
5. Frontana-Uribe et al.
6. Nykaza et al.
7. Mills et al.
8. Fish et al.

1. Intra-molecular cyclization
2. Acylation, nitrosation, cyclization
3. Cu
4. Pd
5.
6. R₃P=O
7. Base
8. Lewis-acid

*Hazardous reagents*
*Metal catalysts*
*High temperatures*
*Organic solvents*

1H-Indazoles
2H-Indazoles

1. 83–90 %
2. 30–86 %

3. 30–96 %
4. 52–60 %
5. 70–85 %
6. 34–98 %
7. 40–90 %
8. 4–83 %

C | This work: An enzymatic synthesis of 1H- and 2H-indazoles

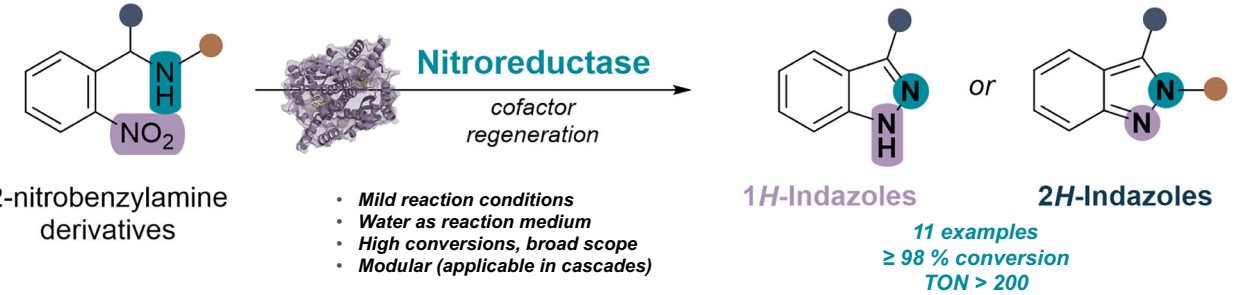

2-nitrobenzylamine derivatives

**Nitroreductase**
*cofactor regeneration*

- *Mild reaction conditions*
- *Water as reaction medium*
- *High conversions, broad scope*
- *Modular (applicable in cascades)*

1H-Indazoles or 2H-Indazoles

*11 examples*
*≥ 98 % conversion*
*TON > 200*

**Fig. 1 | Strategies for indazole formation. A** Examples of drugs containing 1H- (lilac) and 2H-indazole (blue) structures; **B** Examples of traditional synthetic methods for the preparation of 1H-indazoles (lilac)[10,11] and 2H-indazoles (blue)[12–14,16,18,32]. The reactions are shown in the supporting Scheme S1; **C** Herein developed single-step nitroreductase-triggered formation of 1H- and 2H-indazoles.

cascades, e.g., for further diversification of indazoles by enzymatic N-methylation[19]. To date, however, the enzymatic synthesis of indazoles remains an unmet challenge, despite the presence of indazoles in natural products[20–23]. Nitroreductases (NRs) are known to catalyze the reduction of nitro (-NO₂) compounds to amines (-NH₂) via nitroso (-NO) and hydroxylamine (-NHOH) intermediates[24]. The reaction catalyzed by NRs occurs sequentially, which is why in some cases nitroso or hydroxylamine intermediates can be observed[25–30]. Recently, a

photoenzymatic system was developed allowing the NR-catalyzed reaction of aromatic nitro compounds to be directed selectively to amines, azoxy or azo compounds, depending on the reaction conditions[25]. Without the assistance of the photocatalytic system, the nitroso intermediate was found to be most difficult to reduce by the NR BaNTR1 from *Bacillus amyloliquefaciens*, leading to a significant accumulation of this intermediate. Inspired by the possibility of generating reactive nitroso intermediates that can lead to spontaneous

A | Reaction pathway for nitroreductase-triggered indazole formation

B | Investigation of nitroreductase-triggered indazole formation

| Entry | Reaction conditions | Conversion to 6a' [%] |
|:---:|:---:|:---:|
| 1 | Standard – BaNTR1 | 97 |
| 2 | Standard – NfsA | 96 |
| 3 | Standard – BmoNR | 96 |
| 4 | Standard – BpNR | 2 |
| 5 | Thermally inactivated BaNTR1[a] | 5 |
| 6 | No NR | 1[b] |

[a] 15 min, 95 °C.
[b] Impurity of **1a**

**Fig. 2 | Indazole formation using nitroreductases (NR). A** NR-triggered indazole formation from 2-nitrobenzylamine derivatives (**1a**) to 1*H*-indazole (**6a'**) via reduction to the nitroso intermediate **2a**, cyclization to **5a** and aromatization to 2*H*-indazole **6a** before final tautomerization. Possible side products of the NR-catalyzed reaction would be the hydroxylamine intermediates (**3a**) and the diamine (**4a**). **B** Left: investigation of NR-triggered indazole formation. Right: reaction course with all compounds observed in the gas chromatographic analysis of the reaction of **1a** with BaNTR1: **1a** (cyan circles), **6a'** (purple squares). Reaction conditions: 5 mM **1a**, 20 μM NR, 0.5 mM NAD$^+$, 8 μM BmGDH, 50 mM D-glucose, 50 mM NaP$_i$ buffer containing 300 mM NaCl pH 7.0, 1 mL total volume, 25 °C, 1000 rpm, 4 h. The data points represent the mean, and the error bars represent the standard deviation of triplicate experiments (*n* = 3). Source data are provided as a Source Data file.

N-N bond formation[31], we hypothesize that an NR-based reaction could potentially lead to selective indazole formation. This requires the generation of reactive nitrosobenzylamine intermediates that spontaneously form the desired indazole scaffold when the cyclization reaction occurs faster than the enzymatic reduction.

Herein, we report a NR-triggered indazole formation based on the direct two-electron reduction of 2-nitrobenzylamines to reactive nitrosobenzylamine intermediates, which spontaneously cyclize and aromatize to the desired indazoles with excellent conversions (up to >99 %, Fig. 1C). To further demonstrate the synthetic value of this route, we successfully coupled an imine reductase (IRED15) to the NR-triggered indazole formation in a sequential cascade reaction. With this cascade, 2*H*-indazoles were accessible from cheap 2-nitrobenzaldehyde and primary amines with up to 85 % conversion and 68 % isolated yield. This approach not only expands the reaction repertoire of flavin-dependent NRs, but also adds to the available enzyme toolbox for the challenging task of indazole synthesis via ring-closing N-N bond formation in a single enzyme-catalyzed reaction step.

## Results and discussion
### Setting up the NR-triggered indazole reaction
We began our investigations by screening an in-house panel of NRs (BaNTR1 from *Bacillus amyloliquefaciens*, NfsA from *Escherichia coli*, BmoNR from *Bacillus mojavensis*, and BpNR from *Blautia producta*) using 2-nitrobenzylamine (**1a**) as a substrate[25,27]. In the first experiment using BaNTR1, it was found that within 1 hour substrate **1a** was mostly depleted. At the same time, none of the potential products **2a**, **3a**, and **4a** could be identified with gas chromatography-mass spectrometry (GC-MS) analysis. Instead, it was found that 1*H*-indazole (**6a'**) was formed (Figs. 2A and 2B right, Fig. S2). The same was found for NfsA and BmoNR, while BpNR was not active towards **1a** (Fig. 2B left, Figure S3). 2*H*-indazole (**6a**), which is formed according to the proposed mechanism, was not detected due to its quick tautomerization to the more stable 1*H*-form **6a'**[7–9].

Control experiments confirmed that the reaction proceeded solely in the presence of BaNTR1, NfsA and BmoNR (Fig. 2B). In addition, we demonstrated that **2a** cyclizes to indazole by titrating base to the reaction, thereby maintaining a stable pH value. This approach enabled us to monitor the gluconic acid formation from the cofactor regeneration system and the protons released by the reaction itself (Fig. S25 and Supporting Methods). It is known that the cyclization of **2a** can occur spontaneously in an acidic aqueous/methanolic solution[32], and that NRs release their intermediates before the next reduction step[30]. However, we cannot rule out the possibility that cyclization is catalyzed by the enzyme even though cyclization would most likely also occur spontaneously. The turnover number (moles$_{product}$ × moles$_{NR}^{-1}$) of BaNTR1, NfsA and BmoNR was determined to be around 240, aligning well with the generally high catalytic activity of NRs[25,27]. The two most active enzymes, BaNTR1 and NfsA, were used for all subsequent experiments. At the same time, BaNTR1

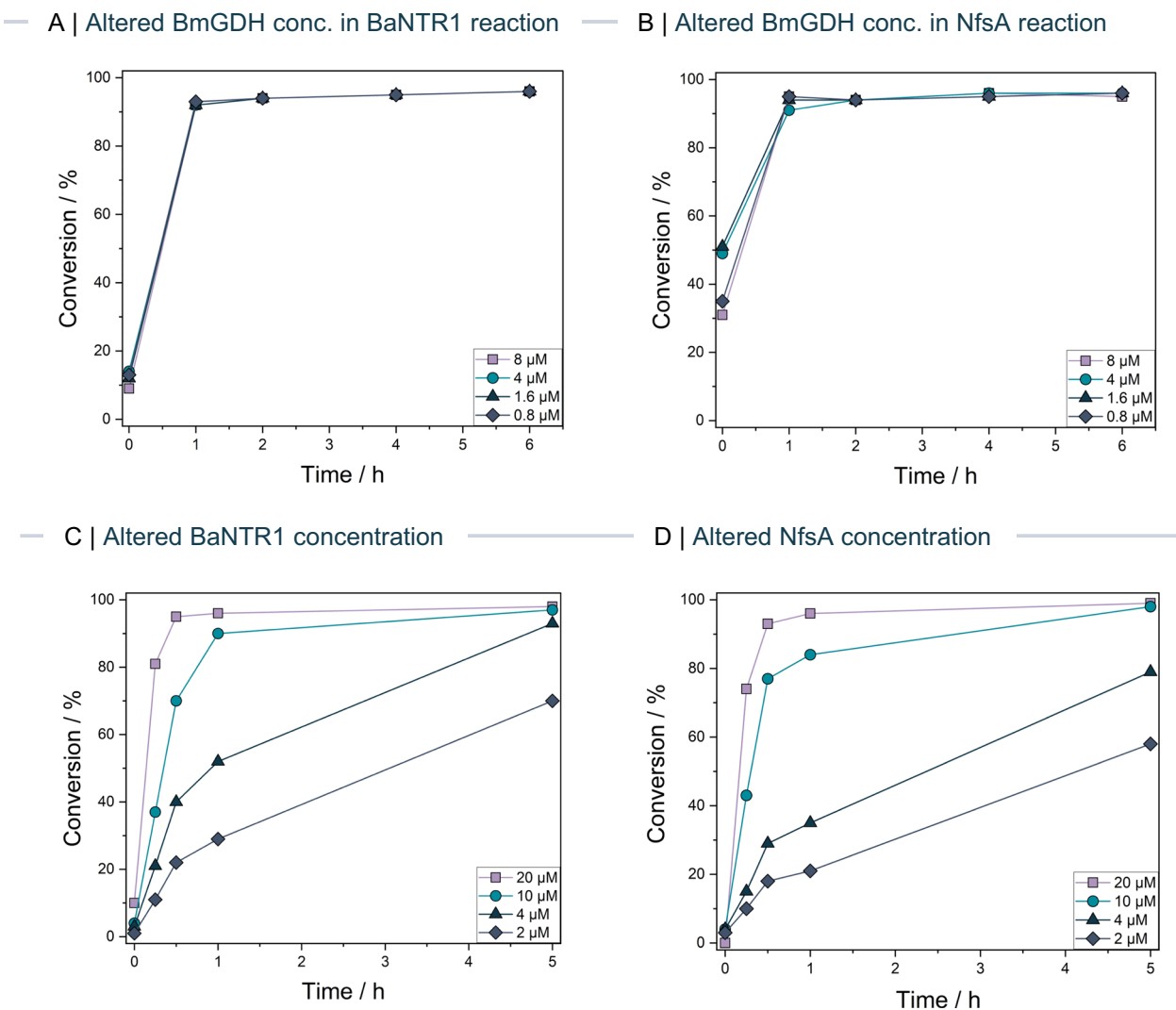

**Fig. 3 | NR and BmGDH reduction experiments for the conversion of** 1a **to** 6a′ **based on gas chromatography (GC) areas.** Standard reaction conditions: 5 mM **1a**, 20 μM NR, 0.5 mM NAD(P)⁺, 8 μM BmGDH, 50 mM D-glucose, 50 mM NaPᵢ buffer + 300 mM NaCl pH 7.0, 1 mL total volume, 25 °C, 1000 rpm, 4 h. **A** Altered BmGDH concentrations in BaNTR1 reaction: Square (purple): 8 μM GDH; circle (cyan): 4 μM GDH; triangle (dark blue): 1.6 μM GDH; diamond (gray): 0.8 μM GDH. **B** Altered BmGDH concentrations in NfsA reaction: Square (purple): 8 μM GDH; circle (cyan): 4 μM GDH; triangle (dark blue): 1.6 μM GDH; diamond (gray): 0.8 μM GDH. **C** Altered BaNTR1 concentrations: Square (purple): 20 μM BaNTR1; circle (cyan): 10 μM BaNTR1; triangle (dark blue): 4 μM BaNTR1; diamond (gray): 2 μM BaNTR1. **D** Altered NfsA concentrations: Square (purple): 20 μM NfsA; circle (cyan): 10 μM NfsA; triangle (dark blue): 4 μM NfsA; diamond (gray): 2 μM NfsA. Source data are provided as a Source Data file.

and NfsA were used as representatives of NADH-dependent and NADPH-dependent NRs, respectively, ensuring that an enzyme with a suitable cofactor preference was available for subsequent cascade development with other oxidoreductases.

## Reaction optimization

Next, we further investigated the NR-triggered conversion of **1a** to **6a′**. First, we investigated whether the NR and glucose dehydrogenase (GDH) concentrations in the reaction with **1a** could be lowered without prolonging the reaction time required for a good conversion (Fig. 3). Reducing the *Bacillus megaterium* GDH (BmGDH) concentration from 8 μM to 0.8 μM had no effect on the course of the reaction, which led us to reduce the BmGDH concentration to 1.6 μM for all further experiments. In contrast, reducing the concentration of NfsA or BaNTR1 resulted in a significant change in the product formation after 1 hour. Considering that some substrates might lead to lower activity when screening the substrate range, the standard concentration of NR was kept at 20 μM.

In addition, we found that the 50 mM phosphate buffer at pH 7.0 used in all previous experiments was not sufficient to prevent the buffer from going into the acidic range during the reaction. This was caused by the gluconic acid formed during the cofactor regeneration cycle. By increasing the buffer strength to 200 mM and raising the pH to 8.0, it was possible to achieve higher or even complete conversion to **6a′** with BaNTR1 and NfsA, respectively (Fig. S4).

To prove the concept, we performed the NR-triggered indazole formation reaction with resting cells and lysate (Figs. S22 and S23). The reactions using lysate reached 96 % conversion after 4 hours, whereas the reactions with resting cells did not exceed 38 % at the same time. This indicates that cofactor regeneration in resting cells or diffusion through the membrane limits the reaction rate. Although whole cell or lysate formulations have advantages for potential industrial applications, we continued using purified enzymes to ensure that any observed biotransformation originates from the NR and not the host background.

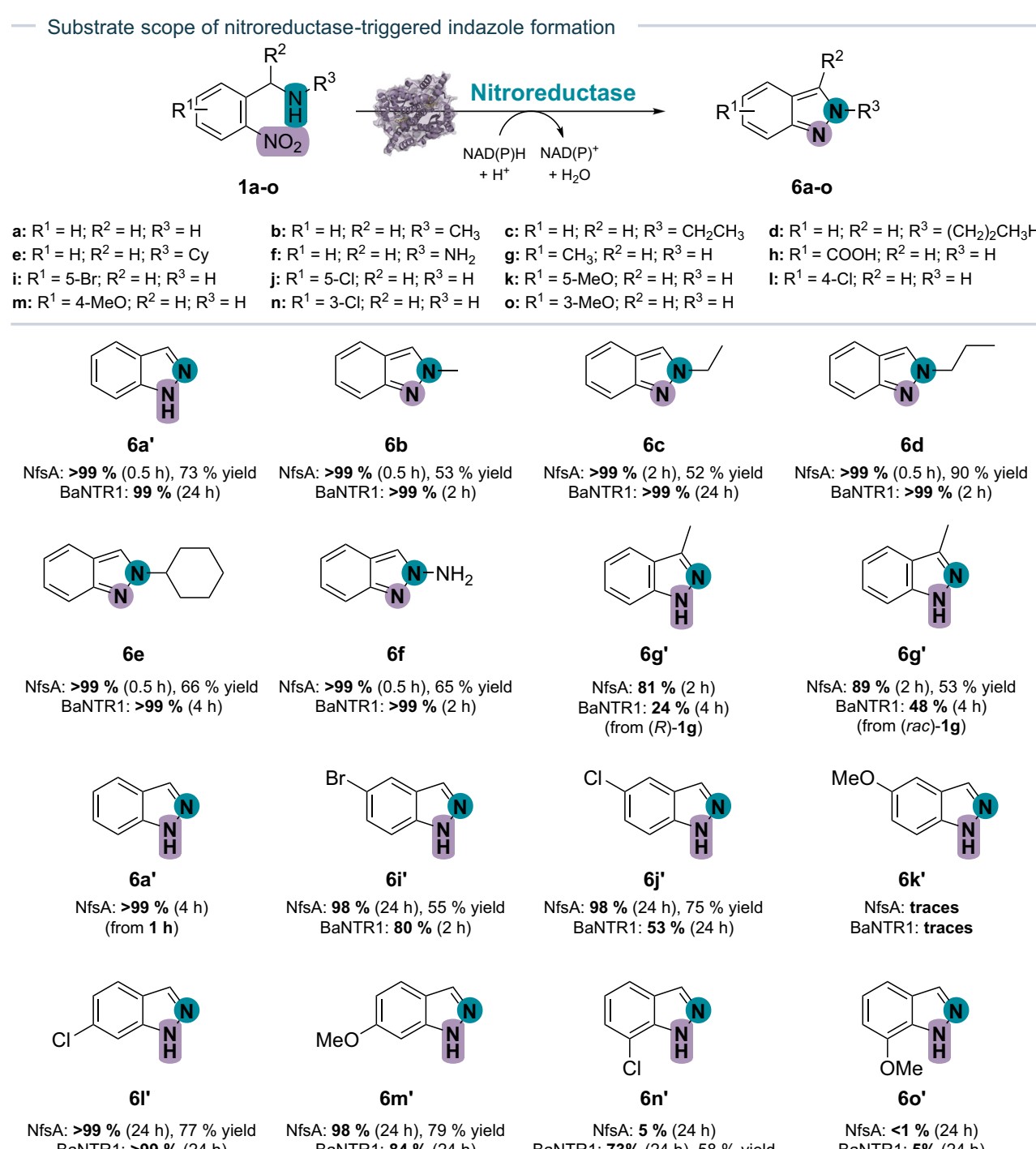

**Fig. 4 | Investigation of the substrate scope of NR-triggered indazole formation.** The conversions based on the GC areas are given for NfsA and BaNTR1, as well as the times required to reach these conversions. Apart from **1n**, isolated yields given were obtained with upscaled reactions using NfsA. In the case of the primary amine substrates (**1a, 1g, 1h, 1i-o**), the 1*H*-indazoles were obtained from the initially generated 2*H*-indazoles, as indicated by an apostrophe. Reaction conditions: 5 mM of substrate, 20 µM NR, 0.5 mM NAD(P)⁺, 1.6 µM BmGDH, 50 mM D-glucose, 200 mM NaPᵢ buffer + 300 mM NaCl pH 8.0, 25 °C, 1000 rpm, 24 h. Reaction volume: analytical scale 1 mL, preparative scale: 20 mL. Source data are provided as a Source Data file.

## Substrate scope analysis

Having established and optimized the NR-triggered indazole formation reaction, we explored the substrate scope of NfsA and BaNTR1 for this transformation (Fig. 4 and Figs. S4–12). A conversion of >99 % was achieved for substrates **1a-f** and **1l** in the case of NfsA. The enzymes generally tolerate polar, non-polar and bulky R³ substitutions, whereas a methyl group on R² strongly reduces activity. Although the indazole products are not chiral, both enzymes seem to prefer the (*S*)-

enantiomer as a substrate, indicated by the fact that the conversion of (*rac*)−**1g** is faster compared to (*R*)−**1g**. Nevertheless, at least in the case of NfsA, conversions of more than 50 % were achieved in the reaction of (*rac*)−**1g**, proving that both enantiomers can be converted. This behavior is advantageous in this particular case because the product of both enantiomers is accessible and thus racemic mixtures can be used. The substrate **1h** did not lead to the expected products, as only the decarboxylated product (**6a'**) was observed in this reaction. It is

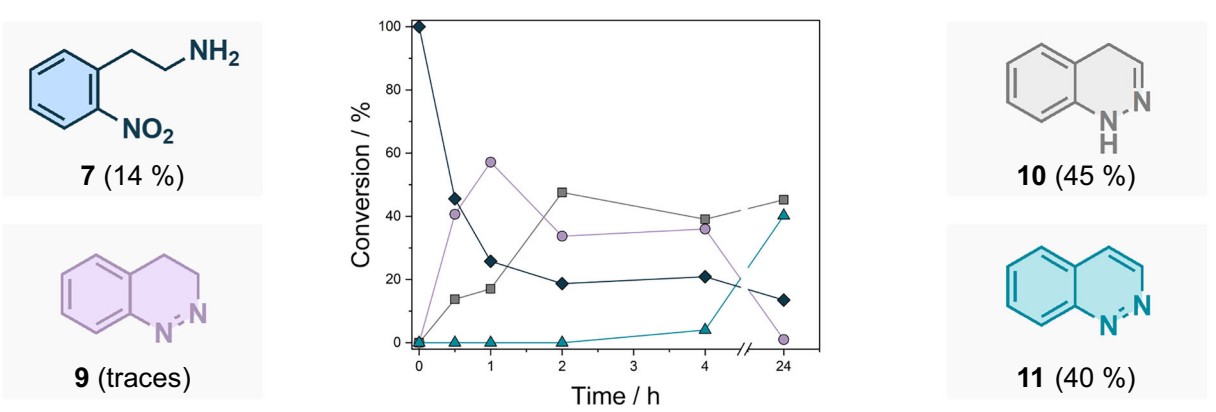

**A | Nitroreductase-triggered cinnoline formation**

**B | Time course for cinnoline formation**

**7 (14 %)**

**9 (traces)**

**10 (45 %)**

**11 (40 %)**

**Fig. 5 | NR-triggered cinnoline formation. A** Reaction scheme of the NR-triggered formation of cinnoline (**11**) via the nitroso intermediate (**8**), 3,4-dihydrocinnoline (**9**) and 1,4-dihydrocinnoline (**10**), mechanism adapted based on literature[33]. **B** Reaction course with all compounds observed in the gas chromatographic analysis of the reaction of **7** with NfsA: **7** (dark blue diamond), **9** (lilac circle), **10** (gray square), and **11** (blue triangle). Reaction conditions: 5 mM **7**, 20 μM NfsA, 0.5 mM NADP⁺, 1.6 μM BmGDH, 50 mM D-glucose, 200 mM NaPi buffer + 300 mM NaCl pH 8.0, 1 mL total volume, 25 °C, 1000 rpm, 24 h. Source data are provided as a Source Data file.

possible that decarboxylation is favored over deprotonation in the aromatization reaction of **5h**. As expected, the $R^3$-substituted substrates lead to 2*H*-indazoles, while the $R^3$-unsubstituted substrates (**1a**, **1g** and **1i-o**) lead to 1*H*-indazoles. For substrates substituted at the aromatic ring (**1i-o**), steric hindrance seems to play an important role. Although NfsA and BaNTR1 accept the 4-chloro- and 4-methoxy-substituted substrates **1l** and **1m**, both enzymes are less active on the 3-chloro- and 3-methoxy-substituted substrates **1n** and **1o**. However, contrary to the general trend, BaNTR1 exhibits higher activity than NfsA with **1n** and **1o**. Interestingly, the 5-bromo- and 5-chloro substituted substrates (**1i** and **1j**) are well converted to the corresponding 1*H*-indazoles. However, when **1k** was used as the substrate, only small amounts of **6k'** were produced and the substrate disappeared overnight. At the same time, the reaction mixture and its extract turned violet; however, no new products were observed on the gas chromatography (GC) chromatogram. This indicates that reactive nitrogen species may have formed oligomers, which is also supported by the observation of precipitate formation over time. This indicates that cyclization is hindered by the strong electron-donating effect of the *para*-methoxy substitution, which reduces the electrophilicity of the nitroso group.

Gas chromatography (GC) and high-resolution mass spectrometry (HR-MS) were performed on all products (Figs. S27–40 and S42–51) to confirm their identity. In addition, the reactions were scaled up to a 20 mg scale for NMR analysis (Figs. S57–79). However, it was found that the products partially evaporated upon prolonged drying in vacuum. Therefore, it was decided to avoid extensive drying of the compounds, which led to solvent residues in the NMR analysis. The isolated yields shown in Fig. 4 were calculated taking into account the remaining solvents (Table S1).

We speculated that the NR-triggered cyclization could be extended towards the formation of 6-membered ring systems, leading us to investigate the formation of cinnoline to further demonstrate the synthetic potential of our strategy. Thus, we used 2-(2-nitrobenzyl)

ethanamine (**7**) to explore the NfsA-triggered cinnoline formation. However, the 3,4-dihydrocinnoline (**9**) formed from the intramolecular condensation of 2-(2-nitrosobenzyl)ethanamine (**8**) must first undergo tautomerization to 1,4-dihydrocinnoline (**10**) and oxidation to the final product cinnoline (**11**)[33]. The last step in particular was not fully completed after 24 hours, resulting in a mixture of 45 % **10** and 40 % **11** with only traces of **9** and 14 % **7** remaining (Fig. 5). In the reaction with BaNTR1, 89 % of substrate **7** remained, while 6 % **10** and 4 % **11** were formed. This experiment demonstrated that cinnolines can also be accessed via the presented NR-triggered route. Overall, NfsA and BaNTR1 showed the same reactivity towards all substrates tested, with NfsA always being more active.

## Kinetic and mechanistic investigations

This synthesis route for indazoles is crucially dependent on enzymatic activity. The direct two-electron reduction of nitro to nitroso compounds can be achieved by NRs. However, the high selectivity of the reaction compared to spontaneous indazole formation is remarkable and the selectivity is in good agreement with our previous observation that the reduction of nitroso intermediates is poorly performed by BaNTR1[25]. The fact that no over-reduction side products **3** and **4** were isolated under the reaction conditions used indicates that the spontaneous cyclization/aromatization of compounds **2** to **6** proceeds rapidly. This is consistent with the finding that nitroso compounds (**2**) were never detected in our reaction mixture. Although the instability of **2** prohibits kinetic studies, the affinity of NfsA and BaNTR1 for **2** might be low, as suggested by kinetic studies with **1a**. The apparent Michaelis-Menten constant ($K_{M,app}$) values for NAD(P)H of $58 \pm 19$ μM and $41 \pm 9$ μM for NfsA and BaNTR1 (Figs. S13 and S15), respectively, are in the typical range for NRs[26,34,35]. However, the $K_{M,app}$ values for **1a** are very high compared to the $K_{M,app}$ value of bacterial nitroreductases with common substrates, such as nitrofurazone, which are below 1 mM[26,36,37]. A $K_{M,app}$ value of $10.8 \pm 2.9$ mM was determined for BaNTR1 (Fig. S15), while the determination for NfsA was not possible because

A | Reaction scheme of the enzymatic cascade reaction

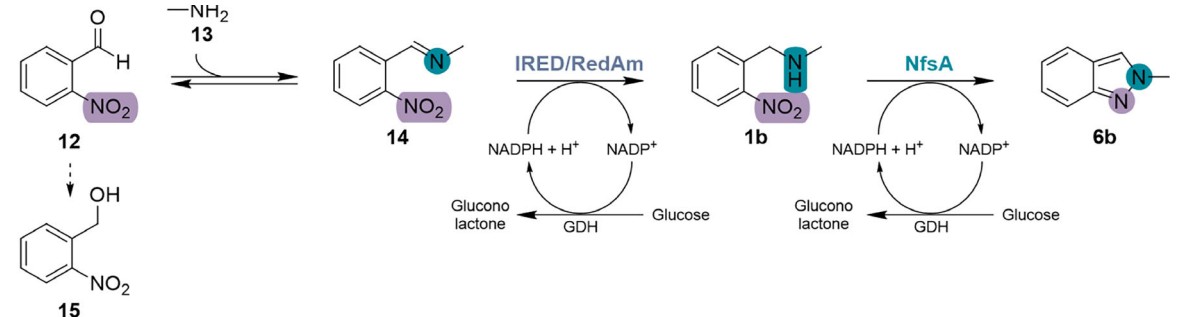

B | IRED screening

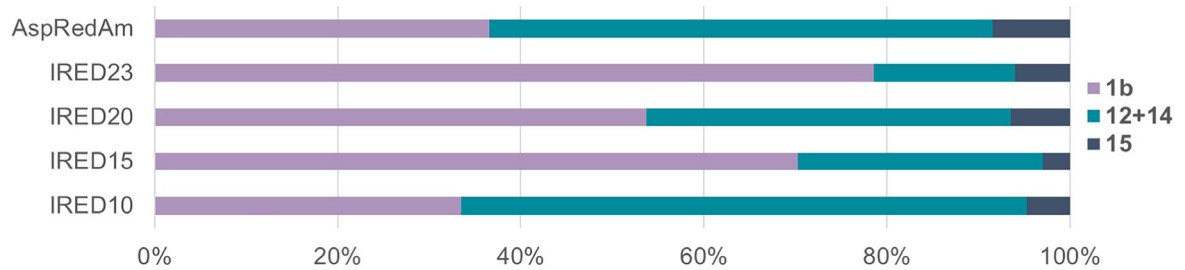

C | Compound distribution of the IRED15 reaction

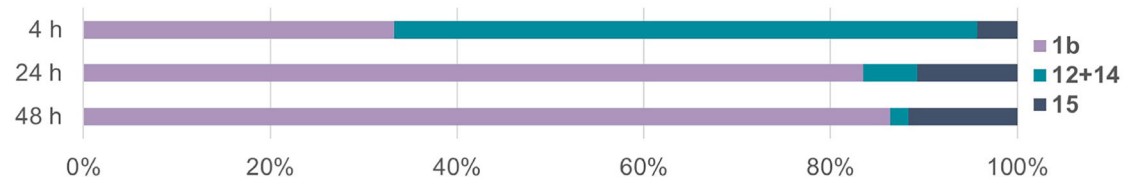

D | Analytical scale cascade IRED15 / NfsA

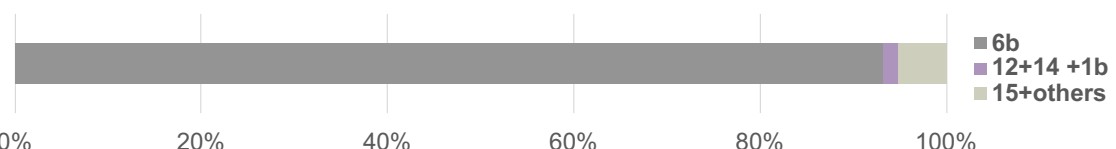

**Fig. 6 | Sequential cascade reaction comprising enzymatic reductive amination and NR-triggered indazole formation. A** Reaction scheme of an enzymatic cascade reaction of IREDs or AspRedAm with NfsA starting from 2-nitrobenzaldehyde (**12**) and methylamine (**13**). The ratio of compounds **12** and **14** changed during the basification process in the sampling, so they were grouped together in the figure. Purple: **1b**; cyan: **12** and **14** together; dark blue: **15**. **B** Composition of compounds obtained after 24 h of reaction of **12** and **13** using AspRedAm and IREDs 10, 15, 20 and 23. Reaction conditions: 5 mM **12**, 20 mM **13**, 20 μM IRED/AspRedAm, 0.5 mM NADP⁺, 50 mM D-glucose, 1.6 μM BmGDH, 50 mM NaPᵢ + 300 mM NaCl + 10 % glycerol pH 8.0, 0.5 mL total volume; 25 °C, 1000 rpm. Purple: **1b**; cyan: **12** and **14** together; dark blue: **15**. **C** Composition of compounds obtained during the reaction of **12** and **13** using IRED15 after 4, 12 and 48 h. Reaction conditions: 5 mM **12**, 20 mM **13**, 40 μM IRED15, 0.5 mM NADP⁺, 50 mM D-glucose, 1.6 μM BmGDH, 50 mM

NaPᵢ + 300 mM NaCl + 10 % glycerol pH 8.0, 0.5 mL total volume; 25 °C, 800 rpm. The ratio of compounds **12** and **14** changed during the basification process in the sampling, so they were grouped together in the figure. **D** Composition of compounds after a sequential analytical scale cascade reaction of IRED15 and NfsA starting from **12** and **13**. The final product of the cascade **6b** is shown separately (gray), while all compounds that could possibly still be converted into the product (**12**, **14**, and **1b**) are grouped together (violet) and side products such as **15** and other partially unidentified peaks, potentially from the nitro reduction of **12, 14** or **15**, are summarized in a light gray bar. Reaction conditions: 5 mM **12**, 20 mM **13**, 40 μM IRED15, 0.5 mM NADP⁺, 50 mM D-glucose, 1.6 μM BmGDH, 50 mM NaPᵢ + 300 mM NaCl + 10 % glycerol pH 8.0, 0.5 mL total volume; 25 °C, 800 rpm. After 48 h, 10 μM NfsA was added and the reaction was incubated for a further 3 h prior to GC analysis. Source data are provided as a Source Data file.

the correlation between indazole formation and substrate concentration was still linear up to 20 mM (Fig. S17). Therefore, the $K_{M,app}$ value of NfsA towards **1a** is more than three orders of magnitude higher than the $K_{M,app}$ value for nitrofurazone (5.5 μM[36]). It was not possible to use higher substrate concentrations in the kinetics, because at substrate concentrations above 20 mM, small amounts of the over-reduced

product **4a** could be detected. These over-reduction products would distort the indazole formation rate. The unusually high $K_{M,app}$ values of NfsA and BaNTR1 for the nitro substrates **1a** in combination with the finding that **4a** formation could only be observed at higher substrate concentrations and thus higher formation of nitroso intermediates suggests that the affinity for the nitroso compounds is also low. This

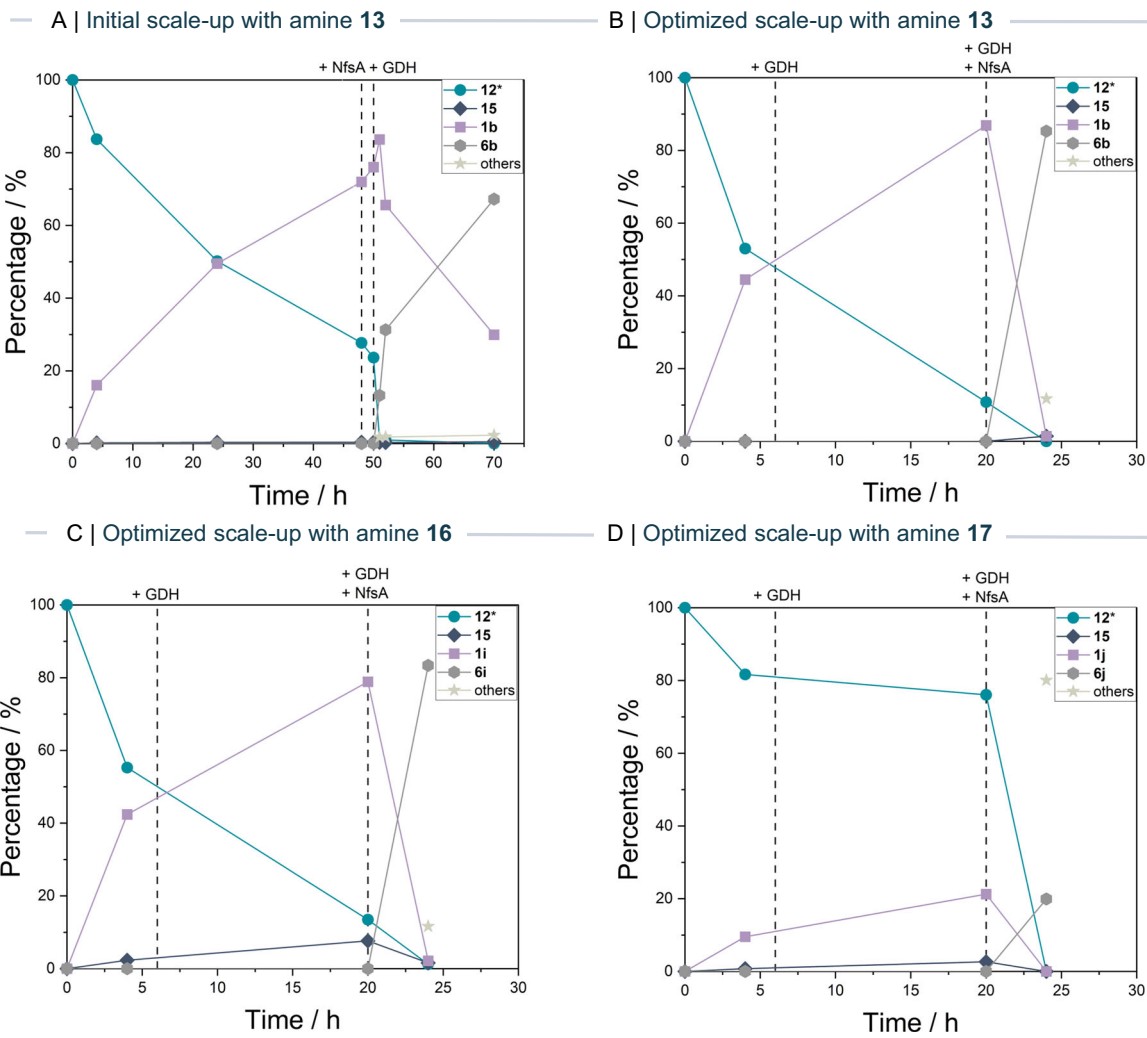

**Fig. 7 | Distribution of compounds during the sequential cascade of IRED15 and NfsA with starting materials** 12 **and** 13, 16 **or** 17. All cascade reactions were carried out on a scale of 50 mg of **12** (about 60 mL reaction volume). Conversions of compounds based on GC areas: **12** (cyan circle), **15** (dark blue diamond), **1b, i, j** (purple square), **6b, i, j** (gray hexagon) and cumulative unidentified side products (light green star). **A** Initial reaction with substrates **12** and **13**. Reaction conditions: 5 mM **12**, 20 mM **13**, 40 μM IRED15, 0.5 mM NADP⁺, 50 mM D-glucose, 1.6 μM BmGDH, 50 mM NaP$_i$ pH 8.0; 25 °C, 180 rpm. After 48 h: 10 μM NfsA was added and 1 h later 1.6 μM BmGDH was added. The reaction then continued for additional 20 h (70 h in total). **B** Optimized reaction with substrates **12** and **13**. Reaction conditions:

5 mM **12**, 20 mM **13**, 20 μM IRED15, 0.5 mM NADP⁺, 50 mM D-glucose, 1.6 μM BmGDH (plus additional doses of the same concentration during the course of the reaction), reaction buffer (50 mM NaP$_i$ + 300 mM NaCl +10 % glycerol, pH 8.0); 25 °C, 180 rpm. After 20 h of reaction time, 10 μM NfsA were added. The reaction was stopped after 24 h. **C** Optimized reaction with substrates **12** and **16**. Same reaction conditions as described in **B**, but using amine **16**. **D** Optimized reaction with substrates **12** and **17**. Same reaction conditions as described in **B**, but using amine **17**. *The percentage of imine was added to the percentage of free substrate **12** because the ratio of these two compounds varies greatly due to the basification performed during sample preparation. Source data are provided as a Source Data file.

explains the complete selectivity towards indazole formation via the spontaneous cyclization/aromatization reaction.

### Cascade reaction–proof-of-concept

One of the major advantages of enzymatic reactions over chemical reactions is that they typically occur under mild aqueous reaction conditions at neutral pH, which also facilitates the development of enzymatic cascade reactions without an intermediate work-up step. Having demonstrated the versatility and efficiency of NR-triggered indazole formation, we decided to further showcase its synthetic utility by developing an enzymatic cascade for the facile synthesis of indazoles from cheap and readily available building blocks. Thus, we envisioned a sequential enzymatic cascade using imine reductases (IREDs) or reductive aminases (RedAms) as well-established biocatalysts to form secondary amines from a carbonyl and a primary amine compound[38–42]. These enzymes could provide the 2-nitrobenzylamines required for the NR-triggered indazole formation from cheap and

readily available 2-nitrobenzaldehyde and a primary amine (Fig. 6). Furthermore, one-pot enzymatic cascades including IREDs and NRs have recently been shown to be highly efficient for synthesizing heterocyclic pharmaceutical building blocks[43]. To prove this concept, IREDs from *Mycobacterium smegmatis* (IRED10), *Actinomadura rifamycini* (IRED15), *Streptomyces tsukubaensis* (IRED20), *Streptomyces viridochromogenes* (IRED23) and the RedAm from *Aspergillus oryzae* (AspRedAm) were investigated for the generation of **1b** from 2-nitrobenzaldehyde (**12**) and methylamine (**13**) (Fig. 6A).

It was found that all selected enzymes were able to convert **12** and **13** into **1b**, with IRED15 and IRED23 showing the highest conversions (Fig. 6B). The reduction of substrate **12** to 2-nitrobenzyl alcohol (**15**) was observed, which is a known side reaction of the IREDs and AspRedAm[40]. For further cascade optimization and upscaling to preparative scale, IRED15 was selected for further experiments. This arose from the fact that the expression of IRED15 at 16.22 μmol/L was much better than the expression of IRED23 at around 0.16 μmol/L. Given the

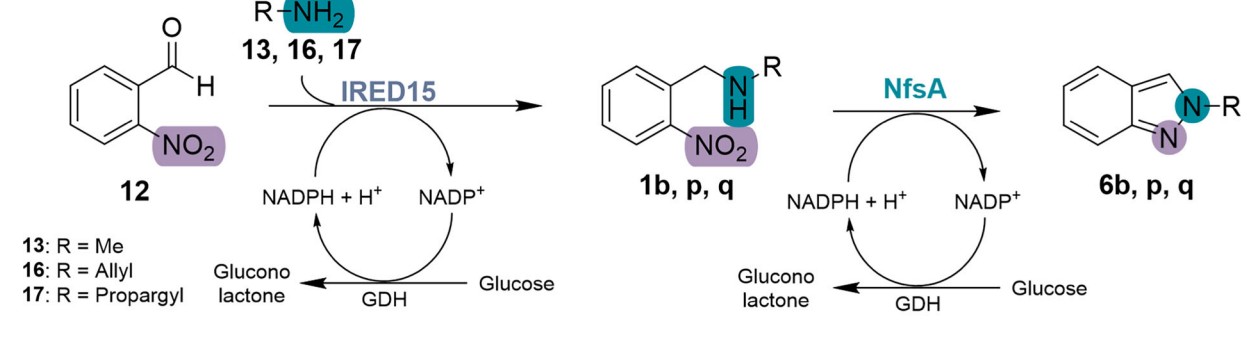

**Substrate scope of the sequential cascade of IRED15 and NfsA**

**6b**
85 % conversion
68 % yield

**6p**
83 % conversion
67 % yield

**6q**
20 % conversion
4 % yield

**Fig. 8 | Cascade reactions on a preparative scale (50 mg of 12, about 60 mL), comprising enzymatic reductive amination and NR-triggered indazole formation.** The enzymatic cascade reactions were performed with IRED15 and NfsA starting from 2-nitrobenzaldehyde (**12**) and different amines: methylamine (**13**), allylamine (**16**) or propargylamine (**17**). Substrates **12** and **13** are converted to **1b** and **6b**, substrates **12** and **16** are converted to **1p** and **6p**, and substrates **12** and **17** are converted to **1q** and **6q**. Source data are provided as a Source Data file.

high expression yield, the concentration of IRED15 was increased to 40 µM. This resulted in over 80 % conversion after 24 h, and 86 % after 48 h (Fig. 6C). When 10 µM NfsA was added to the reaction after 48 h, more than 90 % of **6b** was formed within 3 h (Fig. 6D). The residual percentages mostly consist of small amounts of unidentified side products that may have arisen from the nitroreduction of **12, 14** and **15**.

### Cascade reaction–optimization

During the initial upscaling attempts to the 50-mg scale, the first reaction step was observed to occur much more slowly than in analytical-scale reactions, and the NfsA reaction step began only after further BmGDH was added (Fig. 7A). We concluded that BmGDH was inactivated during the initial reaction step, which caused the slower reaction in the upscaled reactions. Therefore, we investigated the influence of temperature, buffer composition, pH and NaCl concentration on the IRED15-catalyzed conversion of **12** and **13** to **1b**. Interestingly, increasing the temperature from 25 °C to 30 °C decreased the conversion after 24 h, even though IRED15 was previously used at 30 °C[42], again indicating that the stability of the cofactor regeneration system limits the reaction (Fig. S18). Since BmGDH is unstable at higher pH values[44], the reaction was investigated at pH 7.5 (Fig. S18). However, similar conversions to **1b** were achieved, while more side product **15** was formed. Replacing the phosphate-based buffer with tris(hydroxymethyl)aminomethane (Tris) and the cosolvent acetonitrile with dimethyl sulfoxide (DMSO) decreased the reaction conversions (Fig. S19). Although high NaCl concentrations are expected to increase BmGDH stability[44], only an increase in the formation of side product **15** was observed (Fig. S20). To overcome the limitations of the BmGDH-based cofactor regeneration system, we replaced it with a cofactor recycling system based on a mutant formate dehydrogenase (FDH) from *Candida boidinii* (FDH-M4)[45]. Similar conversions were achieved in the IRED15 reaction after 24 h (Fig. S21). However, due to the comparably high concentrations of FDH-M4 required (62 µM), we decided to continue with the BmGDH-based

cofactor regeneration system. Nevertheless, the linear progression of the reaction using the FDH-M4-mediated cofactor regeneration indicates that the cofactor regeneration system is rate-limiting. Thus, for the upscaling reaction at a scale of 50 mg, the IRED15 concentration was reduced to 20 µM, and an additional 1.6 µM dose of BmGDH was added after 6 h of reaction time and before the addition of NfsA (after 20 h). Furthermore, to avoid air bubbles that had been observed in the shake-flask, the reactions were split into three 50 mL reaction tubes.

Using this optimized system, 87 % conversion to **1b** was achieved after the first 20-h reaction step, and 85 % conversion to the final product (**6b**) was achieved after a total reaction time of 24 h with an isolated yield of 68 % (30 mg, Figs. 7B, 8).

### Cascade reaction–extended substrate scope

To showcase the substrate scope of the cascade reaction, we performed the same reaction with allylamine (**16**) and propargylamine (**17**) as the amine substrate instead of **13**. While 83 % overall conversion (67 % isolated yield, 35 mg) was achieved with **16** (Figs. 7C, 8), only 20 % conversion (4 % isolated yield, 2 mg) was reached with **17** due to observed protein precipitation (Figs. 7D, 8). In all these cascades, NR-triggered indazole formation was shown to be highly robust, completely converting the available substrates (**1a, 1p** and **1q**) within a 4-h reaction time. This underlines the potential of NR-triggered indazole formation in cascade reactions.

Overall, we demonstrated that an IRED-NR cascade can efficiently be used in a modular synthesis approach for indazoles, in which the nitrogen atoms in the nitrogen-nitrogen bond do not originate from the same molecule. This not only avoids the use of metal catalysts and harsh conditions, but also the use of nitrogen-nitrogen bonded building blocks such as hazardous hydrazine derivatives or azides. Thus, the biocatalytic route reported herein not only expands the existing synthetic toolbox, but also represents an alternative to chemical and electrochemical methods, overcoming sustainability concerns and providing access to indazole derivatives.

## Methods

### Protein expression

The enzymes BaNTR1 (*Bacillus amyloliquefaciens*, UniProt accession number: A0AAW3IFH5), NfsA (*Escherichia coli*, UniProt accession number: P17117), BmoNR (*Bacillus mojavensis*, UniProt accession number: A0AAP3FXD2), and BpNR (*Blautia producta*, UniProt accession number: A0A7G5MPE8) were expressed[25]. *E. coli* BL21(DE3) cells harboring the pET21a plasmid encoding the NR gene were grown in 1 L of 2xYT medium supplemented with 100 µg/mL ampicillin (100 mg/mL stock) at 37 °C. Upon reaching an $OD_{600}$ of 0.6–0.8, protein expression was induced with 0.3 mM IPTG (1 M stock), and the culture was incubated overnight at 18 °C. Cells were harvested by centrifugation, resuspended, and lysed by sonication. The resulting clarified cell lysate (CFL) was incubated with Ni-NTA agarose beads for 3 h at 4 °C with gentle rotation. The mixture was then loaded onto a gravity flow column, and the bound protein was eluted using appropriate buffers. To replenish any FMN cofactor lost during purification, the eluted enzyme was incubated with 1 mM FMN (10 mM stock) for 3 h at 4 °C with gentle rotation. Finally, the purified enzyme was desalted and buffer-exchanged using PD-10 desalting columns (GE Healthcare).

The expression of FDH-M4, IRED10, IRED15, IRED20, IRED23 and AspRedAm was performed on the basis of a previously described protocol[42,45]. The proteins were expressed in *E. coli* BL21(DE3) strains containing pET16-, pET22- or pET28-based plasmids with the genes of FDH-M4, IREDs or AspRedAm, respectively. Cultures were grown at 37 °C (180 rpm) in 1 L *lysogeny* broth (LB) containing 100 µg/mL ampicillin (100 mg/mL stock) or 50 µg/mL kanamycin ampicillin (50 mg/mL stock). Prior to induction at an $OD_{600}$ of 0.6, the culture was cooled to 20 °C. After induction with 0.2 mM isopropyl β-D-1-thiogalactopyranoside (1 M stock) the culture was further incubated overnight (20 h) at 20 °C. After harvesting, the proteins were lysed by sonication and purified using the His-tag. The elution buffer (50 mM sodium phosphate, 300 mM sodium chloride, 250 mM imidazole, pH 8.0) was replaced with a storage buffer (50 mM sodium phosphate with 300 mM sodium chloride, 10 % glycerol, pH 8.0) using PD-10 desalting columns (GE Healthcare, Chicago, IL, USA).

### Substrate scope screening

All solutions were prepared in 200 mM sodium phosphate buffer (NaP$_i$) with 300 mM sodium chloride pH 8.0, unless otherwise stated. The total volume of the reactions was 1 mL and 1.5 mL reaction tubes were used. 20 µM NR (NfsA: 650 µM stock; BaNTR1 325 µM stock), 1.6 µM BmGDH (640 µM stock), 50 mM D-glucose (1 M stock) and 0.5 mM NAD$^+$ or NADP$^+$ (10 mM stock) were used. The reaction was started by adding 5 mM substrate **1a-o** or **7** from a 10 mM stock solution. Only **1 d** was added from a 200 mM stock solution in acetonitrile. For solubilization, acetonitrile was added to the reactions to a final concentration of 2.5 % (*v/v*), and incubated for 24 h at 25 °C and 1000 rpm. Time samples of 100 µL were taken after 0, 0.5, 1, 2, 4 and 24 h. The time samples were quenched by adding a spatula of potassium phosphate (K$_3$PO$_4$) and then extracting three times with 100 µl ethyl acetate. For substrates **1b-g** and **1i-n**, 2 mM 1*H*-indazole was added to ethyl acetate used as an internal standard. The organic phases were combined and dried over magnesium sulfate before analysis by gas chromatography (GC, see Supplementary information). The conversions of analytical scale reactions are calculated based on the integrals of the GC-FID analysis. Preparative reactions (20 mg scale) for substrates **1a-g, 1i, 1j** and **1l-n** were performed (see Supplementary Information).

**2-Nitrobenzylamine (1a).** 0.150 mmol (28 mg) of **1a·HCl** were used as a starting material and 0.110 mmol (13 mg, 73 %) of product **6a'** were obtained as a colorless solid. $^1$H NMR (400 MHz, CDCl$_3$) δ 10.51 (s, 1H), 8.11 (s, 1H), 7.78 (d, *J* = 8.1 Hz, 1H), 7.51 (d, *J* = 8.4 Hz, 1H), 7.40 (t, *J* = 7.7 Hz, 1H), 7.18 (t, *J* = 7.5 Hz, 1H). GC-MS (EI, 70 eV): *m/z* = 118 (M$^+$, 100), 91 (31), 63 (41), 52 (19). Furthermore, the identity was confirmed by using an authentic standard.

**N-Methyl-2-nitrobenzylamine (1b).** 0.099 mmol (20 mg) of **1b·HCl** were used as a starting material and 0.053 mmol (7 mg, 53 %) of product **6b** were obtained as a colorless solid. $^1$H NMR (400 MHz, CDCl$_3$) δ 7.89 (s, 1H), 7.69 (d, *J* = 8.7 Hz, 1H), 7.64 (d, *J* = 8.4 Hz, 1H), 7.29 (d, *J* = 7.5 Hz, 1H), 7.08 (t, *J* = 7.6 Hz, 1H), 4.22 (s, 3H). $^{13}$C NMR (101 MHz, CDCl$_3$) δ 149.1, 126.2, 123.8, 122.1, 121.9, 120.1, 117.2, 40.4. GC-MS (EI, 70 eV): *m/z* = 132 (M$^+$, 100), 104 (24), 63 (24), 51 (18). HRMS (ESI) *m/z* for C$_8$H$_8$N$_2$ [M + H]$^+$: expected: 133.0760, found: 133.0758.

**N-Ethyl-2-nitrobenzylamine (1c).** 0.107 mmol (23 mg) of **1c·HCl** were used as a starting material and 0.055 mmol (8 mg, 52 %) of product **6c** were obtained as a colorless solid. $^1$H NMR (400 MHz, CDCl$_3$) δ 7.93 (d, *J* = 1.0 Hz, 1H), 7.74–7.70 (m, 1H), 7.65 (dt, *J* = 8.5, 1.1 Hz, 1H), 7.28 (ddd, *J* = 8.7, 6.6, 1.1 Hz, 1H), 7.08 (ddd, *J* = 8.4, 6.6, 0.9 Hz, 1H), 4.49 (q, *J* = 7.3 Hz, 2H), 1.64 (t, *J* = 7.3 Hz, 3H). $^{13}$C NMR (101 MHz, CDCl$_3$) δ 148.8, 126.0, 122.0, 121.7, 120.2, 117.4, 48.6, 16.0. GC-MS (EI, 70 eV): *m/z* = 146 (M$^+$, 40), 118 (100), 91 (20), 63 (29), 51 (15). HRMS (ESI) *m/z* for C$_8$H$_8$N$_2$ [M + H]$^+$: expected: 147.0917, found: 147.0915.

**N-Propyl-2-nitrobenzylamine (1d).** 0.101 mmol (20 mg) of **1 d** were used as a starting material and 0.091 mmol (15 mg, 90 %) of product **6 d** were obtained as a colorless solid. $^1$H NMR (400 MHz, CDCl$_3$) δ 7.91 (d, *J* = 1.0 Hz, 1H), 7.71 (dq, *J* = 8.8, 0.9 Hz, 1H), 7.65 (dt, *J* = 8.4, 1.1 Hz, 1H), 7.33–7.23 (m, 1H), 7.08 (ddd, *J* = 8.3, 6.6, 0.9 Hz, 1H), 4.38 (t, *J* = 7.1 Hz, 2H), 2.10–1.99 (m, 2H), 0.95 (t, *J* = 7.4 Hz, 3H). $^{13}$C NMR (101 MHz, CDCl$_3$) δ 148.9, 125.9, 122.8, 121.8, 121.7, 120.2, 117.4, 55.5, 24.1, 11.3. GC-MS (EI, 70 eV): *m/z* = 160 (M$^+$, 46), 131 (92), 118 (100), 91 (25), 77 (23), 63 (22), 51 (22). HRMS (ESI) *m/z* for C$_8$H$_8$N$_2$ [M + H]$^+$: expected: 161.1072, found: 161.1073.

**N-Cyclohexyl-2-nitrobenzylamine (1e).** 0.068 mmol (21 mg) of **1e·HBr** were used as a starting material and 0.045 mmol (9 mg, 66 %) of product **6e** were obtained as a colorless solid. $^1$H NMR (400 MHz, CDCl$_3$) δ 7.95 (d, *J* = 1.0 Hz, 1H), 7.72 (dd, *J* = 8.8, 1.1 Hz, 1H), 7.65 (dt, *J* = 8.5, 1.1 Hz, 1H), 7.33–7.21 (m, 1H), 7.15–6.99 (m, 1H), 4.49–4.33 (m, 1H), 2.34–2.18 (m, 2H), 2.05–1.83 (m, 4H), 1.85–1.71 (m, 1H), 1.59–1.39 (m, 2H), 1.41–1.27 (m, 1H). $^{13}$C NMR (101 MHz, CDCl$_3$) δ 148.3, 125.7, 121.5, 120.3, 120.2, 117.5, 63.0, 34.1, 25.6, 25.4. GC-MS (EI, 70 eV): *m/z* = 200 (M$^+$, 23), 131 (13), 118 (100), 91 (16), 55 (23). HRMS (ESI) *m/z* for C$_8$H$_8$N$_2$ [M + H]$^+$: expected: 201.1386, found: 201.1382.

**(2-Nitrobenzyl)hydrazine (1f).** 0.100 mmol (20 mg) of **1f·HCl** were used as a starting material and 0.065 mmol (9 mg, 65 %) of product **6 f** were obtained as a colorless solid. $^1$H NMR (400 MHz, CDCl$_3$) δ 7.99 (s, 1H), 7.63 (t, *J* = 9.2 Hz, 2H), 7.34 (dd, *J* = 8.8, 6.7 Hz, 1H), 7.12 (t, *J* = 7.6 Hz, 1H), 4.28 (s, 4H). $^{13}$C NMR (101 MHz, CDCl$_3$) δ 145.4, 127.3, 123.2, 122.4, 120.5, 120.5, 116.3. GC-MS (EI, 70 eV): *m/z* = 133 (M$^+$, 87), 118 (14), 104 (100), 78 (34), 63 (46), 51 (68). HRMS (ESI) *m/z* for C$_8$H$_8$N$_2$ [M + H]$^+$: expected: 134.0713, found: 134.0713.

**1-(2-Nitrophenyl)ethan-1-amine (1g).** 0.100 mmol (20 mg) of **1g·HCl** were used as a starting material and 0.053 mmol (7 mg, 53 %) of product **6g'** were obtained as a colorless solid. $^1$H NMR (400 MHz, CDCl$_3$) δ 7.69 (d, *J* = 8.1 Hz, 1H), 7.49 (d, *J* = 8.4 Hz, 1H), 7.41 (t, *J* = 7.6 Hz, 1H), 7.17 (t, *J* = 7.5 Hz, 1H), 2.65 (s, 3H). $^{13}$C NMR (101 MHz, CDCl$_3$) δ 142.7, 140.8, 127.9, 122.3, 121.0, 120.5, 110.4, 11.8. GC-MS (EI, 70 eV): *m/z* = 132 (M$^+$, 100), 104 (20), 77 (21), 63 (30), 51 (39). HRMS (ESI) *m/z* for C$_8$H$_8$N$_2$ [M + H]$^+$: expected: 133.0760, found: 133.0759.

**5-Bromo-2-nitrobenzylamine (1i).** 0.100 mmol (27 mg) of **1i·HCl** were used as a starting material and 0.055 mmol (11 mg, 55 %) of product **6i'** were obtained as a colorless solid. $^1$H NMR (400 MHz, CDCl$_3$) $\delta$ 8.05 (s, 1H), 7.92 (dd, $J$ = 1.7, 0.8 Hz, 1H), 7.48 (dd, $J$ = 8.8, 1.8 Hz, 1H), 7.41 (dd, $J$ = 8.7, 0.9 Hz, 1H). $^{13}$C NMR (101 MHz, CDCl$_3$) $\delta$ 138.8, 134.2, 130.2, 125.0, 123.5, 114.3, 111.4. GC-MS (EI, 70 eV): $m/z$ = 198 (78), 196 (M$^+$, 83), 117 (69), 90 (100), 63 (65). HRMS (ESI) $m/z$ for C$_8$H$_8$N$_2$ [M + H]$^+$: expected: 196.9705, found: 196.9709.

**5-Chloro-2-nitrobenzylamine (1j).** 0.100 mmol (22 mg) of **1j·HCl** were used as a starting material and 0.075 mmol (11 mg, 75 %) of product **6j'** were obtained as a colorless solid. $^1$H NMR (400 MHz, CDCl$_3$) $\delta$ 8.05 (s, 1H), 7.75 (d, $J$ = 1.9 Hz, 1H), 7.45 (d, $J$ = 8.7 Hz, 1H), 7.36 (dd, $J$ = 8.9, 1.9 Hz, 1H). $^{13}$C NMR (101 MHz, CDCl$_3$) $\delta$ 138.5, 134.5, 127.8, 126.8, 121.5, 120.3, 111.0. GC-MS (EI, 70 eV): $m/z$ = 152 (M$^+$, 100), 125 (35), 90 (35), 63 (43). HRMS (ESI) $m/z$ for C$_8$H$_8$N$_2$ [M + H]$^+$: expected: 153.0214, found: 153.0214.

**4-Chloro-2-nitrobenzylamine (1l).** 0.100 mmol (22 mg) of **1 l·HCl** were used as a starting material and 0.077 mmol (12 mg, 77 %) of product **6 l'** were obtained as a yellowish solid. $^1$H NMR (400 MHz, CD$_3$OD) $\delta$ 8.04 (d, $J$ = 1.7 Hz, 1H), 7.74 (dd, $J$ = 8.6, 0.9 Hz, 1H), 7.55 (dd, $J$ = 1.8, 0.9 Hz, 1H), 7.12 (dd, $J$ = 8.6, 1.7 Hz, 1H). $^{13}$C NMR (101 MHz, CD$_3$OD) $\delta$ 140.7, 133.7, 132.7, 121.8, 121.6, 121.5, 109.5. GC-MS (EI, 70 eV): $m/z$ = 152 (M$^+$, 100), 125 (33), 90 (38), 63 (47). HRMS (ESI) $m/z$ for C$_8$H$_8$N$_2$ [M + H]$^+$: expected: 153.0214, found: 153.0214.

**4-Methoxy-2-nitrobenzylamine (1m).** 0.100 mmol (22 mg) of **1m·HCl** were used as a starting material and 0.079 mmol (12 mg, 79 %) of product **6m'** were obtained as a colorless solid. $^1$H NMR (400 MHz, CDCl$_3$) $\delta$ 7.99 (s, 1H), 7.61 (d, $J$ = 8.9 Hz, 1H), 6.89−6.80 (m, 2H), 3.87 (s, 3H). $^{13}$C NMR (101 MHz, CDCl$_3$) $\delta$ 159.9, 141.5, 134.8, 121.7, 118.0, 113.5, 90.7, 55.6. GC-MS (EI, 70 eV): $m/z$ = 148 (M$^+$, 100), 133 (84), 105 (47), 78 (18). HRMS (ESI) $m/z$ for C$_8$H$_8$N$_2$ [M + H]$^+$: expected: 149.0709, found: 149.0709.

**3-Chloro-2-nitrobenzylamine (1n).** 0.100 mmol (22 mg) of **1n·HCl** were used as a starting material and 0.058 mmol (9 mg, 58 %) of product **6n'** were obtained as a colorless solid. $^1$H NMR (400 MHz, CDCl$_3$) $\delta$ 8.20 (s, 1H), 7.69 (dd, $J$ = 8.0, 0.8 Hz, 1H), 7.40 (dd, $J$ = 7.5, 0.8 Hz, 1H), 7.13 (dd, $J$ = 7.8, 7.8 Hz, 1H). $^{13}$C NMR (101 MHz, CDCl$_3$) $\delta$ 138.3, 135.8, 126.2, 124.8, 122.0, 119.7, 115.9. GC-MS (EI, 70 eV): $m/z$ = 152 (M$^+$, 100), 125 (17), 117 (20), 90 (27), 63 (17). HRMS (ESI) $m/z$ for C$_8$H$_8$N$_2$ [M + H]$^+$: expected: 153.0214, found: 153.0214.

### Kinetics

For the kinetic studies of BaNTR1 and NfsA towards **1a**, a similar biocatalytic setting was used as for the substrate scope screening. The enzyme concentration and reaction time were reduced to 1 μM and 0.4 μM and 45 min and 30 min for BaNTR1 and NfsA, respectively. Also, no acetonitrile was added and the reaction volume was reduced to 0.2 mL. The concentrations of **1a** between 0.5 mM and 25 mM and 2.5 mM to 20 mM were investigated for BaNTR1 and NfsA, respectively. No cofactor regeneration system was used for the kinetic studies of NADH for BaNTR1 and NADPH for NfsA. The concentration of the enzyme was lowered to 0.2 μM and the reaction time was set to 10 min. NADH/NADPH concentrations between 0 and 500 μM and 1 mM of **1a** were used. Kinetics parameters were calculated using OriginPro2020 (OriginLab Corporation, Northampton, MA, USA) with a nonlinear Michaelis-Menten function and an orthogonal distance regression.

### IRED and RedAm screening

In order to find a suitable enzyme for the formation of **1b** from **12** and **13**, AspRedAm and the IREDs 10, 15, 20, and 23 were investigated. In a 1.5 mL reaction tube, 0.5 mL reaction mixture consisting of 5 mM **12**

(200 mM stock in acetonitrile), 20 mM **13** (200 mM aqueous stock, pH 8), 0.5 mM NADP$^+$ (10 mM stock), 50 mM D-glucose (1 M stock), 1.6 μM BmGDH, 20 μM IRED/AspRedAm were prepared in 50 mM sodium phosphate buffer (pH 8.0) with 300 mM NaCl and 10 % ($v/v$) glycerol. The reaction was started by adding the IREDs or AspRedAm and incubated for 24 h at 25 °C and 1000 rpm in a shaker (ThermoMixer C, Eppendorf SE, Hamburg, Germany). 100 μL of the reaction mixture was added to another reaction tube containing 200 μL ethyl acetate and 50 μL saturated potassium phosphate solution (K$_3$PO$_4$). The reaction was vortexed, centrifuged and the organic phase dried over magnesium sulfate prior to gas chromatographic analysis (see Supplementary Information).

### Cascade reaction

The analytical-scale cascade reaction of IRED15 and NfsA was performed in a sequential manner. First, a reaction of 0.5 mL was prepared as in the IRED screening described above, but using 40 μM IRED15. After 4, 24, and 48 h, samples of 100 μL were taken and analyzed as described above. After analyzing the 48 h-time samples, 10 μM NfsA was added and incubation was continued for a further 3 h. A sample of 100 μL was analyzed using the method described above.

The cascade reaction of IRED15 and NfsA was scaled up to the preparative scale (50 mg). For this purpose, 3.31 mL 1 M D-glucose solution, 3.31 mL 10 mM NADP$^+$ solution, 333 μL of 1 M **12** (in acetonitrile, 0.331 mmol, 50 mg), 6.62 mL amines **13, 16**, or **17** (200 mM aqueous stock at pH 8.0), 918 μL acetonitrile and 43.6 mL reaction buffer (50 mM sodium phosphate buffer, 300 mM NaCl, 10 % glycerol, pH 8.0) were added to a 200 mL Erlenmeyer flask. After mixing, the reaction of about 60 mL was divided into three equal portions and incubated in 50 mL reaction tubes. The pH was checked before the reaction started. Then 6.62 mL (3 × 2.21 mL) IRED15 (400 μM stock, 20 μM final conc.) was added. 495 μL (3 × 165 μL) of BmGDH solution (228 μM stock, 1.6 μM) was added at 0, 6 and 20 h. After 20 h 945 μL NfsA (700 μM stock, 10 μM final conc.) was added. The reaction was incubated at 25 °C and 180 rpm (New Brunswick Innova42, Eppendorf SE, Hamburg, Germany) for overall 24 h. Time samples were taken after 4, 20, and 24 h. Each reaction tube was extracted five times with 10 mL of methyl *tert*-butyl ether (MTBE). The combined organic phases were evaporated in the presence of silica and purified by medium-performance liquid chromatography (MPLC). The MPLC separation was performed using a Büchi MPLC system (C-815 Flash, Flawil, Switzerland) equipped with a Flashpure EcoFlex silica column (4 g, 50 μm irregular). The mobile phase consisted of *n*-heptane and MTBE. Gradient elution began with 5 % MTBE, which was maintained for 5 min, followed by a gradual increase in MTBE concentration to 50 % over 30 min. The flow rate was set to 20 mL/min. Detection was performed using a UV detector at 254 nm. The fractions containing the desired product were combined and dried with a rotary evaporator.

The cascade reaction described above already utilizes the optimized reaction conditions. In comparison, the initial cascade was carried out in a 200-mL shaking flask, with a reaction time of 48 h for the first step and 24 h for the second step. During the first reaction step, 40 μM IRED15 was used. The second step was supposed to start with the addition of 10 μM NfsA, but the final product **6b** only formed after the addition of another 1.6 μM BmGDH.

**Methylamine (13).** 30 mg (0.224 mmol, 68 %) of product **6b** were obtained as a colorless solid. $^1$H NMR (400 MHz, CDCl$_3$) $\delta$ 7.89 (d, $J$ = 0.9 Hz, 1H), 7.70 (dq, $J$ = 8.7, 0.9 Hz, 1H), 7.64 (dt, $J$ = 8.4, 1.1 Hz, 1H), 7.28 (ddd, $J$ = 8.7, 6.6, 1.1 Hz, 1H), 7.08 (ddd, $J$ = 8.3, 6.7, 0.9 Hz, 1H), 4.22 (s, 3H). $^{13}$C NMR (101 MHz, CDCl$_3$) $\delta$ 149.1, 126.2, 123.8, 122.1, 121.9, 120.1, 117.2, 40.4. GC-MS (EI, 70 eV): $m/z$ = 132 (M$^+$, 100), 104 (24), 63 (24), 51 (18). HRMS (ESI) $m/z$ for C$_8$H$_8$N$_2$ [M + H]$^+$: expected: 133.0760, found: 133.0758.

**Allylamine (16).** 35 mg (0.222 mmol, 67 %) of product **6p** were obtained as a yellowish oil. [1]H NMR (400 MHz, CDCl$_3$) $\delta$ 7.93 (d, $J$ = 1.1 Hz, 1H), 7.72 (dq, $J$ = 8.7, 1.0 Hz, 1H), 7.65 (dt, $J$ = 8.5, 1.1 Hz, 1H), 7.32–7.24 (m, 1H), 7.08 (ddd, $J$ = 8.4, 6.6, 0.9 Hz, 1H), 6.20–6.06 (m, 1H), 5.38–5.26 (m, 2H), 5.04 (dt, $J$ = 6.1, 1.4 Hz, 2H). [13]C NMR (101 MHz, CDCl$_3$) $\delta$ 149.0, 132.4, 126.1, 122.7, 122.1, 121.8, 120.2, 119.6, 117.6, 56.2. GC-MS (EI, 70 eV): $m/z$ = 158 (M$^+$, 77), 131 (100), 118 (29), 104 (47), 77 (59), 73 (30), 63 (66), 51 (40). HRMS (ESI) $m/z$ for C$_8$H$_8$N$_2$ [M + H]$^+$: expected: 159.0917, found: 159.0914.

**Propargylamine (17).** 2 mg (0.012 mmol, 4 %) of product **6q** were obtained as a yellowish oil. [1]H NMR (400 MHz, CDCl$_3$) $\delta$ 8.19 (d, $J$ = 1.0 Hz, 1H), 7.70 (dq, $J$ = 8.8, 1.0 Hz, 1H), 7.67 (dt, $J$ = 8.4, 1.1 Hz, 1H), 7.30 (ddd, $J$ = 8.8, 6.6, 1.1 Hz, 1H), 7.09 (ddd, $J$ = 8.4, 6.6, 0.9 Hz, 1H), 5.26 (dd, $J$ = 2.6, 0.4 Hz, 2H), 2.61 (t, $J$ = 2.6 Hz, 1H). [13]C NMR (101 MHz, CDCl$_3$) $\delta$ 149.1, 126.7, 122.8, 122.2, 121.9 120.4, 117.4, 76.1, 75.9, 43.1. GC-MS (EI, 70 eV): $m/z$ = 156 (M$^+$, 48), 129 (31), 102 (34), 76 (30), 73 (100), 63 (64), 57 (27), 51 (29). HRMS (ESI) $m/z$ for C$_8$H$_8$N$_2$ [M + H]$^+$: expected: 157.0760, found: 157.0757.

## Reporting summary

Further information on research design is available in the Nature Portfolio Reporting Summary linked to this article.

## Data availability

Data relating to the materials and methods, experimental procedures, mechanistic studies, standard curves, response factors, enzyme kinetics, gas chromatography spectra, HRMS spectra, and NMR spectra are available in the Supplementary Information. All raw data are available from the corresponding author upon request. Protein accession codes: A0AAW3IFH5, P17117, A0AAP3FXD2, A0A7G5MPE8. Source data are provided with this paper.

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

## Acknowledgements

This work was supported by a NWO-VIDI grant from the Netherlands Organization for Scientific Research (NWO, VI.Vidi.213.025) and the European Research Council, ERC (grant agreement number 101075934, ReCNNSTRCT). We also acknowledge financial support from the Netherlands Organization for Scientific Research (VICI grant 724.016.002 awarded to G.J.P.). We thank Francesco G. Mutti (University of Amsterdam) for kindly providing the plasmids encoding IREDs and AspRedAm and Dirk Tischler (Ruhr-University Bochum) for kindly providing the plasmids encoding FDHs. We would also like to thank Ronald van Merkerk (University of Groningen) for developing the pH-STAT device used for mechanistic investigations.

## Author contributions

H.T., L.M., and E.C. designed and performed most of the experiments. A.P.L. and G.J.P. contributed NRs and knowledge about these enzymes. H.T. and S.S. conceived the project, and S.S. directed it. H.T. wrote the manuscript and all authors jointly edited the manuscript. All authors approved the final manuscript.

## Competing interests

The authors declare no competing interests.
