## [Transparent Peer Review file · Nature Communications]

Nitroreductase-triggered indazole formation

Corresponding Author: Professor Sandy Schmidt

Version 1:

Reviewer comments:

Reviewer #1

(Remarks to the Author)

The manuscript submitted by Schmidt and co-workers describes the development of a biocatalytic method for transforming ortho-nitrobenzylamine derivatives into indazoles, using nitroreductase-catalysed reduction of the substrates' nitro group into a nitroso group as the key step. Extension of the method to ortho-nitrophenethylamine, and hence to cinnoline formation, and coupling of the reaction to an upstream IRED-catalysed reductive amination that forms nitrobenzylamines in situ from nitrobenzaldehydes and primary amines are also demonstrated. As pointed out by the authors in the introduction to their article, indazoles are frequent structural motifs in pharmaceutical drugs, but current synthesis methods often require problematic reagents. The new method presented in this manuscript (and, in particular, the two-step cascade) is of considerable interest as it uses simple starting materials and forms the target compounds cleanly and in high analytical yields.

The reported research has been carried out competently, and with only a few minor exceptions the presented data firmly support the authors' conclusions. The manuscript is well structured and written, and the experimental procedures – again with few minor exceptions – are described in sufficient detail to allow them to be reproduced by others in the field. The reported reactions are conceptually and synthetically interesting, and the authors have characterised and optimised them adequately. The demonstrated substrate scope is a bit narrow, in that most N-substituents are simple alkyl groups, the only C-alpha substituents are methyl and carboxyl, and additional substituents on the aromatic ring have not been explored at all. However, this is acceptable for a first proof-of-concept publication on this novel biocatalytic synthesis method.

Overall, this submission describes innovative, high-quality research that will appeal to a broad readership and is therefore suitable for publication in Nature Communications. My only minor concerns are related to product quantification, as well as terminology and formatting issues, and I trust that they can be dispelled with a moderate additional effort. Hence, my recommendation is Minor Revision. Suggestions for improving the manuscript are provided below:

(1) Product quantification: The authors state in the captions of Figures 3 and 4 that the reported conversion values are based on GC areas. However, the way in which the conversions were calculated is not made entirely clear: Are they based on raw integrals, integrals corrected by response factors, or integrals corrected by calibration curves based on an internal standard? What is the role of the internal standards used in the analyses (toluene for GC-MS, 1a for GC-FID)? If the conversions are based on raw (uncorrected) integrals, I consider this acceptable for GC-FID analysis of the target reaction, as substrate and product (nitrobenzylamine and indazole) are likely to have a similar FID response. However, I consider it problematic for the cascade, as the responses of the additional compounds involved there (aldehyde, alcohol, imine) could differ substantially from those of nitrobenzylamine and indazole.

Moreover, as the target reaction proceeds through several (mostly non-enzymatic) steps and via highly reactive intermediates that might undergo side reactions, it would be desirable to perform quantification based on an internal standard, so that recoveries (mass balances) can be reported in addition to conversions. This would also help clarify whether the moderate isolated yields at complete conversion are due to the small scale of the preparative experiments or due to side reactions. All that said, I have to commend the authors for carrying out preparative biotransformations for all reported substrates.

(2) Terminology: On page 4, the authors refer to $\text{TON} = \text{moles product} \times \text{moles NR-1}$ as the "total turnover number". However, the total turnover number is usually abbreviated TTN and is calculated as the molar quantity of product relative to catalyst *over the entire lifetime of the catalyst* (see <https://doi.org/10.1021/cs3005264> and <https://doi.org/10.1002/cite.202200177> for discussions of TON and TTN). As the reactions under question reach complete conversion within a few hours, at which time the enzymes are likely still active, the reported number is an underestimation of the maximum ("total") number of turnovers that can be achieved under these conditions, and should be referred to as "turnover number".

What the authors refer to as "by-products" (e.g., alcohol 15 in the cascade process) are actually side products, as they reduce the yield of the desired compounds (for a discussion of these terms, see: <https://doi.org/10.1021/op300317g>). Please revise.

On several occasions in the main article and the SI, the authors use the wording "were dried" to refer to the removal of solvent by rotary evaporation. This is misleading, as the term "drying" usually refers to the removal of residual water from an organic solution by drying agents such as anhydrous salts. I suggest to use "were evaporated" or "were concentrated" instead.

(3) Formatting: In Figure 4, I suggest to use superscript rather than subscript numbers to distinguish between the two variable residues (R1 and R2), as a subscript R2 can be misread as "two times R". In Figure S2, the colours mentioned in the caption do not always agree with those used in the graph and the legend.

Please ensure correct and consistent formatting (lowercase 3 in CDCl₃, italic J and δ) in all NMR listings, including the reference comparisons, and report ¹³C chemical shifts with one decimal place. Please ensure consistent formatting of journal titles in the SI references.

(4) p. 4, Figure 2 caption: The wording "Error bars represent the standard deviation of measurements made in triplicate." suggests that only the analysis was performed three times; however, I assume that each experimental condition (setup of the biotransformation and analysis) was replicated. Please use a wording that is clear in this regard, for instance "Error bars represent the standard deviation of triplicate experiments." For consistency, please add such a statement to all other graphs showing replicate experiments as well.

(5) p. 7/8: "However, it was found that the products partially evaporated upon prolonged drying in vacuum." With boiling points >200 °C, this problem should have been manageable with careful evaporation. Nevertheless, calculating corrected yields from the solvent-containing NMR samples is an acceptable approach. It would be informative, though, to report the actual weights of the samples along with their corrected weights in the SI.

(6) p. 9, discussion of kinetic experiments, and Figures S12–S15: The y-axis of the Michaelis–Menten plots is drawn in units of GC area. The plots would be more informative if the y-axes were drawn in units of apparent activity (1/s for apparent k_{cat}, or U/mg for apparent specific activity). Also, the vastly different value ranges for Figures S12 and S14 vs Figures S13 and S15 seem to indicate that the data points represent vastly different levels of product concentration (and, hence, conversion). The validity of a kinetic analysis based on a non-continuous assay (here: GC) strongly depends on the level of conversion in the individual samples. Please ensure that all samples taken into consideration are representative of initial reaction rates (ideally <15% conversion) and clearly report the range of conversion values observed in these samples.

(7) p. 11: "This arose from the fact that the expression of IRED15 at 16.22 μ mol/L was much better..." I assume this refers to μ mol of pure protein obtained per litre of culture medium. The reported 16.22 μ mol/L would then correspond to roughly 5 g of pure protein per litre of culture, which seems like a very high expression level. Are the numbers correct?

(8) p. 13: "Using this optimized system, 87 % conversion to 1b was achieved after the first 20-hour reaction step, and 85 % conversion to the final product (6b) was achieved after a total reaction time of 24 hours with an isolated yield of 68 % (30 mg, Figure 7B and 8)." I suggest to move this sentence to the end of the previous section.

(9) Methods section and SI: Please specify the concentrations of stock solutions used for all experimental procedures (for instance, in the way you do for the paragraph on "Investigation of the IRED15/BmGDH system" in the SI). Also, please specify the potassium phosphate species that is added during work-up, for instance by giving the formula in parentheses: "potassium phosphate (K₃PO₄)".

(10) p. 15/16: For all substrates that were used in the form of hydrohalides, please indicate the type of salt with the compound abbreviation (e.g., 1a·HCl, 1e·HBr). Otherwise, the specified weights would appear not to correspond to the specified molar quantities.

Instead of repeating the statement about how NMR analysis was performed ("The proton nuclear magnetic resonance (¹H-NMR) analysis was performed...") for every compound, better use the space for providing listings of the NMR, GC-MS and HRMS data.

(11) p. 16, also Figure 4: The reported yield of compound 6d seems to be incorrect. 8 mg of product correspond to 0.050 mmol, a yield of 49%.

(12) SI, p. 3: "using a 400 MHz NMR" is jargon; better write "using a 400 MHz NMR spectrometer". "The spectra were related to CDCl₃..." It is actually the signal of residual CHCl₃ in the deuterated solvent that is used for referencing. Also, it is good practice to explicitly state the chemical shift value of the signal that the spectra were referenced to. Hence, I suggest: "The spectra were reference to the solvent signal (δ = 7.26 ppm)..."

"No ¹³C-NMRs could be measured due to the low amounts of products obtained." With several mg of the compounds available, a few hours of pulsing would have provided good-quality ¹³C NMR spectra. Since all biotransformation products are known compounds, reporting carbon spectra is not critically important, but if the samples are still available and in good condition, I suggest to acquire and add the ¹³C spectra.

(13) SI, p. 3/4; incomplete analysis conditions: For HRMS analysis, please report the ionisation method (I assume ESI) and conditions. For GC-FID and GC-MS analysis, please report the carrier gas and the flow rate or pressure (depending on whether the instrument was operated in constant pressure or constant flow mode). For GC-MS analysis, please also report

the ionisation mode (I assume EI, 70 eV) and the m/z scan range.

(14) SI, NMR spectra: The NMR data are provided in list form, not table form. Please change the Figure captions accordingly.

(15) Typos: Figure 5 caption: Rection -> Reaction; Figure 7 caption: "starting materials 12 and 13" -> "starting materials 12 and 13, 16 or 17"; p. 13: "available substrates (1a, 1i and 1j)" -> "available substrates (1b, 1i and 1j)"; p. 15: product 6a -> product 6a'; p. 17: 918 mL acetonitrile -> 918 μ L acetonitrile; SI, p. 2: intervalls -> intervals; SI, p. 3: "The sample were diluted" -> "The samples were diluted".

Reviewer #2

(Remarks to the Author)

Schmidt and co-workers report an enzymatic strategy for constructing indazole motifs using nitroreductases (NRs), including a coupled NR–imine reductase cascade that delivers 2H-indazoles directly from inexpensive 2-nitrobenzaldehydes. Although numerous synthetic routes to indazoles have been established, biocatalytic indazole construction, even N–N bond formation remains scarcely explored. Accordingly, this study appears to establish the first enzymatic platform for indazole construction and meaningfully broadens the scope of enzyme-mediated heterocycle synthesis while offering a potentially more sustainable alternative to existing chemical methods. The effort devoted to integrating IREDs with NRs to access 2H-indazoles from aldehyde precursors further underscores the synthetic utility of the approach.

When introducing the state-of-the-art, the authors already summarize a variety of organocatalytic and transition-metal-catalyzed strategies. For completeness and to better guide readers from the synthetic community, it would be helpful to discuss the Davis–Beirut indazole synthesis. In addition, some organophosphorus-catalyzed indazole formations also need to be mentioned (e.g., *J. Am. Chem. Soc.* 2017, 139, 6839–6842; *Chem. Eur. J.* 2018, 24, 9090–9100).

Regarding the nitro reduction pathway, the manuscript refers the reduction of nitro groups to amines as “incomplete reduction”. Please specify why this process is incomplete. Because nitroso species are typically more reactive than nitro compounds, a lot of enzymatic reductions can lead to a complete reduction to hydroxylamines or amines. BaNTR1 and NfsA are two of the exceptions. More evidence needs to be provided for the statement “However, the reaction catalyzed by NRs usually leads to xxx”. Relatedly, do you believe indazole formation occurs within the enzyme active site, or after diffusion of a nitroso intermediate into solution? This information could be helpful for readers to design transformations.

Moreover, the authors demonstrate that without enzyme engineering, the transformation works well with purified enzymes. Have the lysate or whole-cell formats for any of the biocatalytic transformation been evaluated? These observations would inform practical adoption and industrial applications.

The substrate scope focusing on side-chain variation is informative. To further demonstrate robustness, please consider including representative ring-substituted examples. Such data would help potential users gauge the method’s tolerance to electronic and steric variations.

The authors also showcased the substrate scope of the nitroreductase-catalyzed indazole formation, different substituents on the side chain were studied. Have the authors tried substituents on the phenyl ring? Those examples could be a good demonstration for a wider audience that this transformation is robust towards different substituents.

The authors also provide careful kinetic analysis, including Michaelis-Menten experiments to obtain the K_M for both NAD(P)H and 1a. The authors mention the K_M value for 1a with BaNTR1 is very high. Please specify comparing to which value, this K_M (10.8–2.9 mM) is very high? Can you also comment on why higher substrate concentration leads to over-reduction products?

Finally, the NR–IRED cascade nicely demonstrates synthetic utility. Cofactor regeneration could be a costly step. Were other regeneration methods tried, such as sodium dithionite, to replace this system?

Some general edits:

1. Please review the References section. All citations need to be in a consistent format. For example, Ref. 9 and 15 need to be fixed.
2. For all quantification data, such as conversions, K_M value etc., it will be helpful to include the calculation methods in the SI. Please add calibration curves and necessary data in the SI.
3. For all compounds not previously reported, ^1H NMR, ^{13}C NMR and HRMS are all needed. Please add.

Reviewer #3

(Remarks to the Author)

This study reports the enzymatic synthesis of indazoles via nitroreductase (NR)-triggered cyclization of 2-nitrobenzylamine derivatives. The authors used NRs to selectively reduce substrates to reactive nitroso intermediates that spontaneously cyclize to form indazoles with high conversions (>99% for most substrates), producing either 1H- or 2H-indazole tautomers depending on substitution patterns. The substrate scope were investigated, and a proof-of-concept cascade combining an imine reductase with NfsA enables one-pot synthesis of 2H-indazoles from inexpensive 2-nitrobenzaldehyde and primary

amines (up to 85% conversion, 68% isolated yield). The work expands biocatalytic strategies for N-N bond formation and provides a new route to indazole scaffolds. However, this reviewer finds the paper not reach Nature Communications in this stage for the following reasons:

1. Although the proposed nitroso intermediate is mechanistically plausible, its formation has not been directly detected, which weakens the mechanistic interpretation. Please provide solid evidence—such as low-temperature NMR, controlled reduction kinetics (e.g., limited BmGDH/NADPH addition), or the use of an N-substituted secondary amine substrate that prevents over-reduction—to trap or visualize the nitroso/hydroxylamine intermediate and strengthen the proposed mechanism.
2. Less than 10 examples were demonstrated with only substitutions on the nitrogen or adjacent carbon. The diversity of the substrate scope is not high. For six membered rings, the scope seems to be too narrow. The extension of the nitroreductase-triggered cyclization to six-membered ring systems (e.g., cinnolines) is conceptually promising but remains underdeveloped. Please optimize the oxidation conditions (e.g., addition of a mild oxidant or use of an enzymatic oxidation system) to achieve full aromatization to cinnoline and expand the substrate scope for six-membered heterocycle formation.
3. Yields are moderate for some cases (6b, 6c, 6e, 6f), and high for 6d, even the conversions were the same. Could further engineering of the enzyme expand the scope of the reactions to include more diverse substrates? Though the reaction seems quite efficient now, the authors should at least attempt enzyme engineering.
4. The two enzymatic steps were performed sequentially, not simultaneously, which should be emphasized.
5. Please cite the recent multienzyme stereodivergent cascade for tetrahydroquinoline synthesis (Angew. Chem. Int. Ed. 2024, 63, e202411561) to better contextualize your cascade strategy within the current literature.
6. Three replicates were mentioned, but no error bars or statistical analyses are provided. Please add mean \pm SD (or SEM) to all bar/line charts and clearly state the sample size (n) and statistical methods used in the figure captions.
7. Please report the total reaction volume (in x mL) for each experiment in the main text.
8. Supply ^{13}C NMR data for all new compounds (6f). Ensure consistent formatting for NMR solvent descriptions throughout the Supporting Information (e.g., "400 MHz, CDCl_3 ").

Version 2:

Reviewer comments:

Reviewer #1

(Remarks to the Author)

In their revised manuscript, Schmidt and co-workers have thoroughly addressed the minor concerns voiced in my original reviewer report, as well as the criticism raised by the other two reviewers. The authors have carried out several additional experiments, the most significant of which are an extension of the substrate scope to ring-substituted derivatives and a titration experiment that supports their original mechanistic proposal. In addition, many small improvements were implemented, ranging from determination of GC response factors to an increased level of detail in figure captions and experimental procedures, and the correction of typos and minor terminological issues.

All these revisions have further strengthened the manuscript, and I recommend its publication in Nature Communications after a last round of minor corrections, which can be implemented without further peer review. Suggestions for these final improvements are provided below:

- (1) Terminology: Please ensure consistent spelling of the term "side product" (with a space, not a hyphen).
- (2) Listings of GC-MS data: It is not common practice to report expected and observed masses in unit-resolution GC-MS analysis. Instead, provide a list of all significant mass fragments including their relative abundance, with identification of the parent ion (M+); e.g., for 6a': GC-MS (EI, 70 eV): m/z = 118 (M+, 100), 91 (31), 63 (40), 52 (19).
- (3) GC-MS analysis conditions: The flow rate of 13.8 mL/min appears unusually high. I assume it is the total flow (including split flow, and septum purge flow, if applicable) that is reported here, but in order to allow other researchers to reproduce the GC separations, the column flow rate (the part of total carrier gas flow that is directed through the column) should be reported.

Reviewer #2

(Remarks to the Author)

The authors have thoroughly addressed all of my comments and concerns, and I have no further objections.

Reviewer #3

(Remarks to the Author)

The authors have comprehensively addressed all the core issues raised in the initial review. The inclusion of additional experimental data, an expanded substrate scope, and enhanced data presentation has substantially improved the scientific rigor, completeness, and overall persuasiveness of the manuscript. My primary concerns have been satisfactorily resolved. Specifically, the newly conducted pH-stat titration experiments provide solid quantitative evidence for the proposed "nitroso-intermediate-mediated cyclization" mechanism. Furthermore, the addition of seven ring-substituted substrates significantly broadens the synthetic utility of the method and allows for a more in-depth discussion of substituent effects. The authors have also meticulously updated the experimental section with essential details, including error bars and reaction volumes, provided the missing NMR data in a consistent format, and updated the relevant citations. This work represents the first

example of nitroreductase-triggered indazole synthesis and demonstrates clear originality. Given its mild reaction conditions, operational simplicity, broad substrate tolerance, and successful integration into a practical enzymatic cascade, I recommend the manuscript for publication.

Response to reviewers' comments

We are grateful to the reviewers for their enthusiastic feedback and helpful suggestions. We have uploaded a revised version of the paper that addresses the reviewers' feedback and suggestions for improving the manuscript. The specific changes made are outlined in the reviewer response file below (author responses are in blue) and highlighted in yellow in the revised manuscript and supporting information. In the reviewer response file below, we also highlighted the changes made to the manuscript and supporting information in yellow. These changes can also be found in the two documents.

Reviewer #1 (Remarks to the Author):

The manuscript submitted by Schmidt and co-workers describes the development of a biocatalytic method for transforming ortho-nitrobenzylamine derivatives into indazoles, using nitroreductase-catalysed reduction of the substrates' nitro group into a nitroso group as the key step. Extension of the method to ortho-nitrophenethylamine, and hence to cinnoline formation, and coupling of the reaction to an upstream IRED-catalysed reductive amination that forms nitrobenzylamines in situ from nitrobenzaldehydes and primary amines are also demonstrated. As pointed out by the authors in the introduction to their article, indazoles are frequent structural motifs in pharmaceutical drugs, but current synthesis methods often require problematic reagents. The new method presented in this manuscript (and, in particular, the two-step cascade) is of considerable interest as it uses simple starting materials and forms the target compounds cleanly and in high analytical yields.

The reported research has been carried out competently, and with only a few minor exceptions the presented data firmly support the authors' conclusions. The manuscript is well structured and written, and the experimental procedures – again with few minor exceptions – are described in sufficient detail to allow them to be reproduced by others in the field. The reported reactions are conceptually and synthetically interesting, and the authors have characterised and optimised them adequately. The demonstrated substrate scope is a bit narrow, in that most N-substituents are simple alkyl groups, the only C-alpha substituents are methyl and carboxyl, and additional substituents on the aromatic ring have not been explored at all. However, this is acceptable for a first proof-of-concept publication on this novel biocatalytic synthesis method.

Overall, this submission describes innovative, high-quality research that will appeal to a broad readership and is therefore suitable for publication in Nature Communications. My only minor concerns are related to product quantification, as well as terminology and formatting issues, and I trust that they can be dispelled with a moderate additional effort. Hence, my recommendation is Minor Revision. Suggestions for improving the manuscript are provided below:

We are very grateful for the reviewer's compliments and detailed suggestions, which have further improved the quality of the manuscript. We greatly appreciate that the reviewer thoroughly read the main manuscript and supplementary information and identified several important aspects that we have carefully addressed during the revision.

(1) Product quantification: The authors state in the captions of Figures 3 and 4 that the reported conversion values are based on GC areas. However, the way in which the conversions were calculated is not made entirely clear: Are they based on raw integrals, integrals corrected by response factors, or integrals corrected by calibration curves based on an internal standard? What is the role of the internal standards used in the analyses (toluene for GC-MS, 1a for GC-FID)? If the conversions are based on raw (uncorrected) integrals, I consider this acceptable for GC-FID analysis of the target reaction, as substrate and product (nitrobenzylamine and indazole) are likely to have a similar FID response. However, I consider it problematic for the cascade, as the responses of the additional compounds involved there (aldehyde, alcohol, imine) could differ substantially from those of nitrobenzylamine and indazole. Moreover, as the target reaction proceeds through several (mostly non-enzymatic) steps and via highly reactive intermediates that might undergo side reactions, it would be desirable to perform quantification based on an internal standard, so that recoveries (mass balances) can be reported in addition to conversions. This would also help clarify whether the moderate isolated yields at complete conversion are due to the small scale of the preparative experiments or due to side reactions. All that said, I have to commend the authors for carrying out preparative biotransformations for all reported substrates.

We are grateful for the reviewers question and we have recapitulated our data. The conversions were actually based on the substrate and product integrals. For clarification, a sentence on this topic has been added to the 'Methods' and the 'Supporting Methods' section.

The conversions of analytical scale reactions are calculated based on the integrals of the GC-FID analysis.

The internal standard was added to track the variation in injection volume of the individual GC runs. We determined the response factors of the compounds in the model cascade yielding **6b** and found that the response of all compounds is fairly similar (not more than 7% difference; Figure S25). Based on the response factors and the total area, the recovery of the cascade reaction presented in Figure 6C and 6D varied between 60 % and 91 % (4 h: 69 %, 24 h: 81 %, 48 h: 91 %, 48h + 3 h NR reaction: 60 %). The fact that the second lowest recovery of 69 % was measured at the first sample (4 h) indicates that the variation in the recovery is mainly caused by variation in the extraction process. Furthermore, all reactions shown in Figures 4 and 8 were

performed on a preparative scale and the isolated yields are given. The difference between conversion and yield can basically be attributed to the small scale and the losses originating from extraction, coating of the compound on silica for the column chromatography, and the column chromatography itself. Being aware of these critical parameters, we managed to lower the product losses in the purification process for most substrates added during the revision to as little as 3 mg.

(2) Terminology: On page 4, the authors refer to $\text{TON} = \text{moles product} \times \text{moles NR-1}$ as the "total turnover number". However, the total turnover number is usually abbreviated TTN and is calculated as the molar quantity of product relative to catalyst *over the entire lifetime of the catalyst* (see <https://doi.org/10.1021/cs3005264> and <https://doi.org/10.1002/cite.202200177> for discussions of TON and TTN). As the reactions under question reach complete conversion within a few hours, at which time the enzymes are likely still active, the reported number is an underestimation of the maximum ("total") number of turnovers that can be achieved under these conditions, and should be referred to as "turnover number".

What the authors refer to as "by-products" (e.g., alcohol 15 in the cascade process) are actually side products, as they reduce the yield of the desired compounds (for a discussion of these terms, see: <https://doi.org/10.1021/op300317g>). Please revise.

On several occasions in the main article and the SI, the authors use the wording "were dried" to refer to the removal of solvent by rotary evaporation. This is misleading, as the term "drying" usually refers to the removal of residual water from an organic solution by drying agents such as anhydrous salts. I suggest to use "were evaporated" or "were concentrated" instead.

We thank the reviewer for these valuable clarifications regarding terminology. We appreciate the distinction between turnover number (TON) and total turnover number (TTN) and agree that our original wording may lead to misunderstanding. As suggested, we revised the manuscript to refer to the reported values as "turnover numbers," thereby avoiding any implication that these represent the maximum turnovers achievable over the full catalyst lifetime.

We also thank the reviewer for pointing out the misuse of the term "by-products." We revised the manuscript accordingly and consistently refer to unintended reaction products such as alcohol **15** as "side products".

Finally, we appreciate the reviewer's observation regarding the phrase "were dried." We agree that this wording is misleading in the context of solvent removal by rotary evaporation. We corrected this phrasing throughout the main text and the Supporting Information, replacing it with "were evaporated".

(3) Formatting: In Figure 4, I suggest to use superscript rather than subscript numbers to

distinguish between the two variable residues (R1 and R2), as a subscript R2 can be misread as "two times R". In Figure S2, the colours mentioned in the caption do not always agree with those used in the graph and the legend.

Please ensure correct and consistent formatting (lowercase 3 in CDCl₃, italic J and δ) in all NMR listings, including the reference comparisons, and report ¹³C chemical shifts with one decimal place. Please ensure consistent formatting of journal titles in the SI references.

We thank the reviewer for the helpful suggestions regarding formatting and figure presentation. We revised Figure 4 and used superscript indices for R¹, R² and R³ to avoid any potential misinterpretation. We also appreciate the reviewer's note concerning Figure S2; we carefully checked the colors in the graph, legend, and caption to ensure full consistency.

Furthermore, we thoroughly checked and corrected all NMR data to ensure uniform formatting, including the use of lowercase "3" in CDCl₃, italic formatting for J and δ , and reporting of all ¹³C chemical shifts with one decimal place. We also thank the reviewer for pointing out inconsistencies in the formatting of journal titles in the Supporting Information; these were corrected to achieve a uniform style throughout.

Example:

¹H NMR (400 MHz, CDCl₃) δ 7.93 (d, J = 1.0 Hz, 1H), 7.74 – 7.70 (m, 1H), 7.65 (dt, J = 8.5, 1.1 Hz, 1H), 7.28 (ddd, J = 8.7, 6.6, 1.1 Hz, 1H), 7.08 (ddd, J = 8.4, 6.6, 0.9 Hz, 1H), 4.49 (q, J = 7.3 Hz, 2H), 1.64 (t, J = 7.3 Hz, 3H).

Reference comparison^[5]: ¹H NMR (400 MHz, CDCl₃): δ 7.90 (s, 1 H), 7.72–7.70 (dd, J = 8.8, 0.8 Hz, 1H), 7.64 (d, J = 8.4 Hz, 1H), 7.29–7.25 (m, 1H), 7.09–7.05 (m, 1H), 4.47 (q, J = 7.2 Hz, 2H), 1.63 (t, J = 7.2 Hz, 3H).

Figure S59. ¹H NMR spectrum/list of **6c** enzymatically produced from **1c** and an NMR list of a reference.

(4) p. 4, Figure 2 caption: The wording "Error bars represent the standard deviation of measurements made in triplicate." suggests that only the analysis was performed three times; however, I assume that each experimental condition (setup of the biotransformation and analysis) was replicated. Please use a wording that is clear in this regard, for instance "Error bars represent the standard deviation of triplicate experiments." For consistency, please add such a statement to all other graphs showing replicate experiments as well.

We thank the reviewer for this helpful remark regarding the wording in the caption of Figure 2. As suggested, we revised the caption to read "Error bars represent the standard deviation of triplicate experiments," which more accurately reflects that each

biotransformation setup, including both reaction and analysis, was independently replicated. We have carefully checked the entire manuscript and Supporting Information accordingly.

The data points represent the mean, and the error bars represent the standard deviation of triplicate experiments (n = 3).

(5) p. 7/8: "However, it was found that the products partially evaporated upon prolonged drying in vacuum." With boiling points >200 °C, this problem should have been manageable with careful evaporation. Nevertheless, calculating corrected yields from the solvent-containing NMR samples is an acceptable approach. It would be informative, though, to report the actual weights of the samples along with their corrected weights in the SI.

We agree with the reviewer that we could have carefully evaporated the solvent for most compounds, as we did for the substrates added during this revision. However, when we started the upscaling reactions, we found that product **6b** was evaporating in high vacuum. We therefore had to repeat this upscaling experiment. This is why we proceeded with greater caution in subsequent upscaling reactions, trying not to evaporate the solvent extensively to avoid product loss. Consequently, more solvent remained in some products. The actual weights are now noted in the supplementary information Table S1.

(6) p. 9, discussion of kinetic experiments, and Figures S12–S15: The y-axis of the Michaelis–Menten plots is drawn in units of GC area. The plots would be more informative if the y-axes were drawn in units of apparent activity (1/s for apparent k_{cat} , or U/mg for apparent specific activity). Also, the vastly different value ranges for Figures S12 and S14 vs Figures S13 and S15 seem to indicate that the data points represent vastly different levels of product concentration (and, hence, conversion). The validity of a kinetic analysis based on a non-continuous assay (here: GC) strongly depends on the level of conversion in the individual samples. Please ensure that all samples taken into consideration are representative of initial reaction rates (ideally <15% conversion) and clearly report the range of conversion values observed in these samples.

We thank the reviewer for the comments regarding the kinetic analysis. We have converted the y-axes of all Michaelis–Menten plots to apparent activity units ($k_{cat,app}$) to make the data more informative and comparable (Figures S12-15). Additionally, based on a calibration curve (Figure S16), we confirmed that conversions in all kinetic samples remained below 6 %, even at the lowest substrate concentrations used. The vast difference in area is due to the fact that the $K_{M,app}$ values of NAD(P)H are in the low μM range. When we performed the NAD(P)H kinetics, we stopped the reaction at low conversions (always below 6 % relative to NAD(P)H), resulting in a low μM concentration of the formed product. Meanwhile, the $K_{M,app}$ values for **1a** were much higher, enabling

higher product concentrations (even in the mM range) without exceeding 6 % conversion relative to **1a**. For clarity, the conversions for the kinetics towards **1a** do not have to be relative to NAD(P)H because a cofactor regeneration system was used in the latter experiments, which avoids NAD(P)H depletion. The kinetics towards NAD(P)H were not analyzed using the cofactor regeneration system to avoid influencing the reaction speed at low NADPH concentrations through competitive binding or slow regeneration.

(7) p. 11: "This arose from the fact that the expression of IRED15 at 16.22 $\mu\text{mol/L}$ was much better..." I assume this refers to μmol of pure protein obtained per litre of culture medium. The reported 16.22 $\mu\text{mol/L}$ would then correspond to roughly 5 g of pure protein per litre of culture, which seems like a very high expression level. Are the numbers correct?

Yes, the numbers are correct. Based on the MW of IRED15 of about 31.4 kg/mol (corresponding to 31.4 mg/ μmol) the amount of protein per liter of culture medium is about 500 mg (16.22 $\mu\text{mol/L}$ * 31.4 mg/ μmol). This corresponds to the expression level we obtain from the culture.

(8) p. 13: "Using this optimized system, 87 % conversion to 1b was achieved after the first 20-hour reaction step, and 85 % conversion to the final product (6b) was achieved after a total reaction time of 24 hours with an isolated yield of 68 % (30 mg, Figure 7B and 8)." I suggest to move this sentence to the end of the previous section.

We thank the reviewer for this helpful suggestion. We agree that the sentence in question fits better at the end of the previous section. We adjusted the manuscript accordingly.

(9) Methods section and SI: Please specify the concentrations of stock solutions used for all experimental procedures (for instance, in the way you do for the paragraph on "Investigation of the IRED15/BmGDH system" in the SI). Also, please specify the potassium phosphate species that is added during work-up, for instance by giving the formula in parentheses: "potassium phosphate (K_3PO_4)".

We thank the reviewer for this constructive comment. We have revised both the Methods section and the Supporting Information to specify the concentrations of all stock solutions used. In addition, we have clarified the potassium phosphate species by adding the chemical formula (K_3PO_4) wherever applicable.

Example:

Substrate scope screening

All solutions were prepared in 200 mM sodium phosphate buffer (NaPi) with 300 mM sodium chloride pH 8.0, unless otherwise stated. The total volume of the reactions was 1 mL and 1.5 mL reaction tubes were used. 20 μ M NR (NfsA: 650 μ M stock; BaNTR1 325 μ M stock), 1.6 μ M BmGDH (640 μ M stock), 50 mM D-glucose (1 M stock) and 0.5 mM NAD⁺ or NADP⁺ (10 mM stock) were used. The reaction was started by adding 5 mM substrate **1a-h**, **1i**, **1j**, **1l-n** or **7** from a 10 mM stock solution.

(10) p. 15/16: For all substrates that were used in the form of hydrohalides, please indicate the type of salt with the compound abbreviation (e.g., **1a**·HCl, **1e**·HBr). Otherwise, the specified weights would appear not to correspond to the specified molar quantities.

Instead of repeating the statement about how NMR analysis was performed ("The proton nuclear magnetic resonance (1H-NMR) analysis was performed...") for every compound, better use the space for providing listings of the NMR, GC-MS and HRMS data.

We thank the reviewer for these helpful suggestions. We have indicated the type of hydrohalide salt for all relevant substrates by adding the corresponding notation, ensuring that the mentioned weights and molar quantities are fully consistent.

Furthermore, we have removed the repeated NMR-analysis statements and added NMR, GC-MS, and HRMS data listings instead.

Example:

0.099 mmol (20 mg) of **1b**·HCl were used as a starting material and 0.053 mmol (7 mg, 53 %) of product **6b** were obtained as a colorless solid. ¹H NMR (400 MHz, CDCl₃) δ 7.89 (s, 1H), 7.69 (d, J = 8.7 Hz, 1H), 7.64 (d, J = 8.4 Hz, 1H), 7.29 (d, J = 7.5 Hz, 1H), 7.08 (t, J = 7.6 Hz, 1H), 4.22 (s, 3H). ¹³C NMR (101 MHz, CDCl₃) δ 149.1, 126.2, 123.8, 122.1, 121.9, 120.1, 117.2, 40.4. GC-MS: expected: m/z 132, found: m/z 132. HRMS (ESI) m/z for C₈H₈N₂ [M+H]⁺: expected: 133.0760, found: 133.0758.

(11) p. 16, also Figure 4: The reported yield of compound **6d** seems to be incorrect. 8 mg of product correspond to 0.050 mmol, a yield of 49%.

We are grateful for spotting this mistake. The mg yield mentioned in paragraph **6d** is actually incorrect due to a copy-paste error from the paragraph of **6c**. While addressing comment 5, we also found that the weight of product **6d** (which was measured to be 16 mg) was not corrected by the remaining solvent as the solvent peaks were comparably low. This has now been corrected, resulting in 15 mg of **6d** corresponding to 0.091 mmol and a yield of 90 %. We have carefully checked all the other yields reported in the manuscript.

(12) SI, p. 3: "using a 400 MHz NMR" is jargon; better write "using a 400 MHz NMR spectrometer". "The spectra were related to CDCl₃..." It is actually the signal of residual CHCl₃ in the deuterated solvent that is used for referencing. Also, it is good practice to explicitly state the chemical shift value of the signal that the spectra were referenced to. Hence, I suggest: "The spectra were reference to the solvent signal ($\delta = 7.26$ ppm)...". "No ¹³C-NMRs could be measured due to the low amounts of products obtained." With several mg of the compounds available, a few hours of pulsing would have provided good-quality ¹³C NMR spectra. Since all biotransformation products are known compounds, reporting carbon spectra is not critically important, but if the samples are still available and in good condition, I suggest to acquire and add the ¹³C spectra.

We thank the reviewer very much for the comments on the NMR analysis description. The NMR methods are now described in a separate paragraph in the SI and follow the suggestions. We also followed the advice to measure the ¹³C NMRs using elongated pulsing times, and we were able to acquire the ¹³C spectra that we added to the SI and the main paper (list only).

Nuclear magnetic resonance (NMR) spectroscopy

*The proton nuclear magnetic resonance (¹H-NMR) and carbon nuclear magnetic resonance (¹³C-NMR) analysis were performed in deuterated chloroform (CDCl₃) using a 400 MHz NMR spectrometer (JOEL 400YH, Peabody, MA, USA). Only **6l** was measured in deuterated Methanol (CD₃OD). The spectra were reference to the solvent signal ($\delta = 7.26$ ppm in case of CDCl₃ and $\delta = 3.31$ ppm in case of CD₃OD). If available, the spectra were compared with those reported in the literature.*

(13) SI, p. 3/4; incomplete analysis conditions: For HRMS analysis, please report the ionisation method (I assume ESI) and conditions. For GC-FID and GC-MS analysis, please report the carrier gas and the flow rate or pressure (depending on whether the instrument was operated in constant pressure or constant flow mode). For GC-MS analysis, please also report the ionisation mode (I assume EI, 70 eV) and the m/z scan range.

Indeed, the HRMS ionization mode is ESI. This information was added to the HRMS section in the SI and each HRMS experiment shown in the main paper. Details on the GC-FID and GC-MS methods were also added:

GC-MS

The measurements were performed using helium as the carrier gas at a flow rate of 13.8 mL/min. The split ratio was set to 5, and the MS detector (EI, 70 eV, m/z 50-500) was turned on after 3 minutes.

GC-FID

The measurements were performed using nitrogen (N₂) as a carrier gas at a pressure of 158.7 kPa. The split ratio was set to 5 for the kinetics and to 20.5 for all other measurements.

(14) SI, NMR spectra: The NMR data are provided in list form, not table form. Please change the Figure captions accordingly.

We agree with the reviewer's comment. We adjusted the figure captions accordingly.

(15) Typos: Figure 5 caption: Rection -> Reaction; Figure 7 caption: "starting materials 12 and 13" -> "starting materials 12 and 13, 16 or 17"; p. 13: "available substrates (1a, 1i and 1j)" -> "available substrates (1b, 1i and 1j)"; p. 15: product 6a -> product 6a'; p. 17: 918 mL acetonitrile -> 918 μ L acetonitrile; SI, p. 2: intervalls -> intervals; SI, p. 3: "The sample were diluted" -> "The samples were diluted".

We thank the reviewer for carefully reading the manuscript and noting these typos. All listed errors have been corrected in the manuscript and Supporting Information.

Reviewer #2 (Remarks to the Author):

Schmidt and co-workers report an enzymatic strategy for constructing indazole motifs using nitroreductases (NRs), including a coupled NR–imine reductase cascade that delivers 2H-indazoles directly from inexpensive 2-nitrobenzaldehydes. Although numerous synthetic routes to indazoles have been established, biocatalytic indazole construction, even N–N bond formation remains scarcely explored. Accordingly, this study appears to establish the first enzymatic platform for indazole construction and meaningfully broadens the scope of enzyme-mediated heterocycle synthesis while offering a potentially more sustainable alternative to existing chemical methods. The effort devoted to integrating IREDs with NRs to access 2H-indazoles from aldehyde precursors further underscores the synthetic utility of the approach.

We thank the reviewer for the valuable comments and suggestions, which we carefully considered during our manuscript revision. These comments have further improved the quality of our manuscript.

When introducing the state-of-the-art, the authors already summarize a variety of organocatalytic and transition-metal-catalyzed strategies. For completeness and to better guide readers from the synthetic community, it would be helpful to discuss the Davis–Beirut indazole synthesis. In addition, some organophosphorus-catalyzed

indazole formations also need to be mentioned (e.g., J. Am. Chem. Soc. 2017, 139, 6839–6842; Chem. Eur. J. 2018, 24, 9090–9100).

We appreciate the reviewer's comment, which has strengthened the background and accessibility of the manuscript for the synthetic community. We have expanded the introduction to include a brief discussion of the Davis–Beirut indazole synthesis as well as relevant organophosphorus-catalyzed approaches. We also included a very recent example of Lewis acid-catalyzed 2*H*-indazole formation. These additions help to provide a more complete overview of current synthetic strategies and better contextualize our biocatalytic method within the broader landscape of indazole synthesis.

*Furthermore, organophosphorus-catalyzed pathways for forming 2*H*-indazoles were developed starting from 2-nitrobenzaldehydes and primary amines^{15,16}. 3-Alkoxy-group-functionalized 2*H*-indazoles can be generated via the classical, base-catalyzed Davis-Beirut reaction. This reaction relies on an intramolecular, redox reaction of 2-nitrobenzylamines, followed by the addition of an alcohol at the 3-position of the indazole ring¹⁷. However, a photochemical acids-catalyzed variation has been established that make these products accessible from 2-nitrobenzyl alcohols, primary amines, and alcohols¹⁸. Additionally, a Lewis acid-catalyzed approach for the synthesis of 2*H*-indazoles from imines of 1-(2-Aminophenyl)ethenone derivatives has been published recently¹⁹.*

Regarding the nitro reduction pathway, the manuscript refers the reduction of nitro groups to amines as “incomplete reduction”. Please specify why this process is incomplete. Because nitroso species are typically more reactive than nitro compounds, a lot of enzymatic reductions can lead to a complete reduction to hydroxylamines or amines. BaNTR1 and NfsA are two of the exceptions.

We thank the reviewer for this useful comment, and we agree that the current phrasing might be misleading. By "incomplete reduction," we meant that the 6e⁻ nitro reduction performed by nitroreductases is not a concerted reaction, and the 2e⁻ (nitroso) or 4e⁻ (hydroxylamine) reduction intermediates can sometimes be observed (e.g., Molecules 23(2), 211 (2018); J. Biol. Chem. 289, 15203–15214 (2014); Nat. Commun. 14, 5442 (2023)). This means that the nitroso intermediate exists and can undergo chemical reactions when they occur quickly enough, even though it is typically described as a more reactive nitroreductase substrate. The point was clarified by rephrasing the according paragraph:

The reaction catalyzed by NRs occurs sequentially, which is why in some cases nitroso or hydroxylamine intermediates can be observed²⁶⁻³¹.

More evidence needs to be provided for the statement “However, the reaction catalyzed by NRs usually leads to xxx”.

We acknowledge the reviewer's point that NR reactions do not always lead to nitroso- or hydroxylamine-type intermediates. We rewrote the entire sentence as described above to weaken the statement.

The reaction catalyzed by NRs occurs sequentially, which is why in some cases nitroso or hydroxylamine intermediates can be observed^{26–31}.

Relatedly, do you believe indazole formation occurs within the enzyme active site, or after diffusion of a nitroso intermediate into solution? This information could be helpful for readers to design transformations.

Unfortunately, the nitroso intermediate is too reactive to be detected experimentally. It is also known to cyclize spontaneously in an aqueous methanolic solution (*Tetrahedron*, 54(1998):3197-3206). Therefore, the cyclization does not have to take place in the protein. NRs work according to a ping-pong bi-bi mechanism (*J Biol Chem*. 2014 Apr 4;289(22):15203–15214), the nitroso intermediate must leave the active site of the NR before a further reduction step can occur. However, cyclization could still occur before the intermediate is released into the solution. Therefore, we cannot exclude the possibility that the enzyme is involved in the cyclization; however, the enzyme is most likely not required.

*It is known that the cyclization of **2a** can occur spontaneously in an acidic aqueous/methanolic solution,¹⁴ and that NRs release their intermediates before the next reduction step.³¹ However, we cannot rule out the possibility that cyclization is catalyzed by the enzyme even though cyclization would most likely also occur spontaneously.*

Moreover, the authors demonstrate that without enzyme engineering, the transformation works well with purified enzymes. Have the lysate or whole-cell formats for any of the biocatalytic transformation been evaluated? These observations would inform practical adoption and industrial applications.

We agree that this would be a useful experiment for the industrial application of the reaction. Therefore, we conducted lysate and whole cell experiments. The results are promising and demonstrate the potential application of this reaction in resting cells and lysate. We added a sentence to the main paper mentioning the applicability of the concept in lysate and whole cells. The experiments are depicted in Figures S21 and S22.

To prove the concept, we performed the NR-triggered indazole formation reaction with resting cells and lysate (Figures S21 and S22). The reactions using lysate reached 96 %

conversion after four hours, whereas the reactions with resting cells did not exceed 38 % at the same time. This indicates that cofactor regeneration in resting cells or diffusion through the membrane limits the reaction rate. Although whole cell or lysate formulations have advantages for potential industrial applications, we continued using purified enzymes to ensure that any observed biotransformation originates from the NR and not the host background.

The substrate scope focusing on side-chain variation is informative. To further demonstrate robustness, please consider including representative ring-substituted examples. Such data would help potential users gauge the method's tolerance to electronic and steric variations. The authors also showcased the substrate scope of the nitroreductase-catalyzed indazole formation, different substituents on the side chain were studied. Have the authors tried substituents on the phenyl ring? Those examples could be a good demonstration for a wider audience that this transformation is robust towards different substituents.

We thank the reviewer for this valuable suggestion. We agree that including ring-substituted examples would further demonstrate the robustness and versatility of the method. We conducted experiments with halogen- and methoxy-substituents in the *ortho*, *meta*, and *para* positions relative to the nitro group (substrates **1i-o**, Figure 4). While the enzymes converted most of the substrates very well, *ortho*-substitution seems to lead to reduced activity due to steric hindrance. BaNTR1 appears to tolerate *ortho*-substitutions better than NfsA. Substrate **1k** was depleted by the enzymes; however, cyclization did not occur efficiently, possibly due to the strong electron-donating effect of the *para*-methoxy group, which reduces the electrophilicity of the nitroso group.

For substrates substituted at the aromatic ring (**1i-o**), steric hindrance seems to play an important role. Although NfsA and BaNTR1 accept the 4-chloro- and 4-methoxy-substituted substrates **1l** and **1m**, both enzymes are less active on the 3-chloro- and 3-methoxy-substituted substrates **1n** and **1o**. However, contrary to the general trend, BaNTR1 exhibits higher activity than NfsA with **1n** and **1o**. Interestingly, the 5-bromo- and 5-chloro substituted substrates (**1i** and **1j**) are well converted to the corresponding 1H-indazoles. However, when **1k** was used as the substrate, only small amounts of **6k'** were produced and the substrate disappeared overnight. At the same time, the reaction mixture and its extract turned violet; however, no new products were observed on the gas chromatography (GC) chromatogram. This indicates that reactive nitrogen species may have formed oligomers, which is also supported by the observation of precipitate formation over time. This indicates that cyclization is hindered by the strong electron-donating effect of the *para*-methoxy substitution, which reduces the electrophilicity of the nitroso group.

The authors also provide careful kinetic analysis, including Michaelis-Menten experiments to obtain the K_M for both NAD(P)H and 1a. The authors mention the K_M value for 1a with BaNTR1 is very high. Please specify comparing to which value, this K_M (10.8 ± 2.9 mM) is very high? Can you also comment on why higher substrate concentration leads to over-reduction products?

Typical NR substrates, such as nitrofurazone, have K_M values below 1 mM (e.g. 5.5 μ M for NfsA), whereas the K_M values of NfsA and BaNTR1 are higher. The example of nitrofurazone was mentioned later in the manuscript. To avoid confusion, the paragraph was reorganized and additional citations were added.

The apparent Michaelis-Menten constant ($K_{M,app}$) values for NAD(P)H of 58 ± 19 μ M and 41 ± 9 μ M for NfsA and BaNTR1 (Figures S12 and S14), respectively, are in the typical range for NRS^{27,34,35}. However, the $K_{M,app}$ values for 1a are very high compared to the $K_{M,app}$ value of bacterial nitroreductases with common substrates, such as nitrofurazone, which are below 1 mM^{27,36,37}. A $K_{M,app}$ value of 10.8 ± 2.9 mM was determined for BaNTR1 (Figure S13), while the determination for NfsA was not possible because the correlation between indazole formation and substrate concentration was still linear up to 20 mM (Figure S15). Therefore, the $K_{M,app}$ value of NfsA towards 1a is more than three orders of magnitude higher than the $K_{M,app}$ value for nitrofurazone (5.5 μ M³⁶).

At higher substrate concentrations, over-reduction has been observed, as with higher substrate concentrations, more nitroso intermediates are produced. This increases their effective concentration even though cyclization occurs quickly. Eventually, the nitroso compound concentration becomes high enough that an uncyclized nitroso intermediate occasionally enters the active site and undergoes further reduction.

Finally, the NR-IRED cascade nicely demonstrates synthetic utility. Cofactor regeneration could be a costly step. Were other regeneration methods tried, such as sodium dithionite, to replace this system?

We are grateful for the acknowledgment of the synthetic utility of the designed cascade. Yes, other cofactor regeneration systems were investigated. We successfully regenerated the cofactor using a variant of formate dehydrogenase (FDH) from *Candida boidinii* (FDH-M4), which was designed for NADPH regeneration (Figure S20). However, due to the fact that about 100 times more FDH is required for similar performance compared to GDH, we continued with the GDH-glucose system to establish the cascade reaction. We agree that non-enzymatic regeneration methods or industrial GDHs that are more stable at pH 8 or higher could be used to optimize the system from an economic point of view. However, since this paper describes the first proof of concept of this cascade and cofactor regeneration is not its focus, we believe that exploring this further is beyond its scope and could be done by industrial users facing these issues in all biocatalytic reductions.

Some general edits:

1. Please review the References section. All citations need to be in a consistent format. For example, Ref. 9 and 15 need to be fixed.

We have carefully checked the references, and we believe they align with the Nature Communications guidelines.

2. For all quantification data, such as conversions, KM value etc., it will be helpful to include the calculation methods in the SI. Please add calibration curves and necessary data in the SI.

We thank the reviewer for this helpful suggestion. We have added the calculation methods for all quantification data to the appropriate places. Additionally, we have included the standard curve used for kinetics (Figures S12-16) and the GC response factor in the SI (Figure S25).

Kinetics:

Kinetics parameters were calculated using OriginPro2020 (OriginLab Corporation, Northampton, MA, USA) with a nonlinear Michaelis-Menten function and an orthogonal distance regression.

Conversions:

*The conversions reported in the manuscript were calculated from the raw integrals of the substrate, side-product (where applicable), and product peaks. This approach was chosen because the response factors of the representative compounds **1b**, **6b**, **12**, **14**, and **15** were determined to be very similar (Figure S25).*

$$\text{Conversion (\%)} = 100 \times \frac{\text{Area}_{\text{product}}}{\text{Area}_{\text{substrate}} + \text{Area}_{\text{side-product}} + \text{Area}_{\text{product}}}$$

3. For all compounds not previously reported, ¹H NMR, ¹³C NMR and HRMS are all needed. Please add.

All of the compounds except **6f** and **6q** have already been described in the literature. Nevertheless, we added the ¹H NMR, HRMS and ¹³C NMR spectra for all of the compounds that we isolated, except for 1*H*-indazole and cinnoline, for which commercial standards were available.

Reviewer #3 (Remarks to the Author):

This study reports the enzymatic synthesis of indazoles via nitroreductase (NR)-triggered cyclization of 2-nitrobenzylamine derivatives. The authors used NRs to selectively reduce substrates to reactive nitroso intermediates that spontaneously cyclize to form indazoles with high conversions (>99% for most substrates), producing either 1H- or 2H-indazole tautomers depending on substitution patterns. The substrate scope were investigated, and a proof-of-concept cascade combining an imine reductase with NfsA enables one-pot synthesis of 2H-indazoles from inexpensive 2-nitrobenzaldehyde and primary amines (up to 85% conversion, 68% isolated yield). The work expands biocatalytic strategies for N-N bond formation and provides a new route to indazole scaffolds. However, this reviewer finds the paper not reach Nature Communications in this stage for the following reasons:

1. Although the proposed nitroso intermediate is mechanistically plausible, its formation has not been directly detected, which weakens the mechanistic interpretation. Please provide solid evidence—such as low-temperature NMR, controlled reduction kinetics (e.g., limited BmGDH/NADPH addition), or the use of an N-substituted secondary amine substrate that prevents over-reduction—to trap or visualize the nitroso/hydroxylamine intermediate and strengthen the proposed mechanism.

We thank the reviewer for this insightful comment. We agree that the mechanistic interpretation could be further strengthened by experiments proving that the nitroso intermediate is cyclizing. For this we conducted an experiment where we indirectly measured the turnover of the cofactor regeneration system. We used a pH probe coupled titration to neutralize the proton released from gluconic acid (hydrolyzed gluconolactone), an equimolar by-product formed during cofactor recycling. We also considered that protons are released during the reaction because the amine group of the substrate is mostly protonated at pH 8.0, but the product, 1H-indazole, does not react as a base at this pH value (pK_a of 1.22; T. K. Adler and A. Albert, J. Chem. Soc., 1960, 1794). To determine this, we measured the pK_a value of **1a** and found it to be 8.63. Experimentally, we found that the addition of 1.62 equivalents of NaOH (relative to the substrate) were needed to keep the pH stable (at 73% conversion (GC-FID)). From the theoretic calculations we expected the addition of 1.32 eq. of NaOH for one turnover of the cofactor regeneration system. Considering that flavin-dependent enzymes typically exhibit a side reaction in which reduced flavin cofactors react with O₂ to form H₂O₂, a reaction that we also detected, this result clearly indicates that only one 2e⁻ reduction step occurs. Therefore, we are confident that cyclization occurs from the nitroso intermediate. We added a statement to the main paper and included the titration experiment in the supplementary information (SI Methods and Figure S24).

In addition, we demonstrated that **2a** cyclizes to indazole by titrating base to the reaction, thereby maintaining a stable pH value. This approach enabled us to monitor the gluconic acid formation from the cofactor regeneration system and the protons released by the reaction itself (Figure S24 and Supporting Methods).

pH titration

A pH probe (pH Electrode InLab Expert Pro, Mettler-Toledo, Columbus, OH, USA), a syringe pump (PHD 2000, Harvard apparatus, Holliston, MA, USA; 1 mL SGE[®] borosilicate glass syringes, TRAJAN, Melbourne, Australia), a control unit consisting of a Raspberry Pi 5 (Raspberry Pi, Cambridge, UK) and a Gravity pH Meter V2.0 (DFRobot, Shanghai, China) were used for automatized titration. 0.1 M NaOH was used as a base. For the calculation of the pKa value of **1a**, 12.5 mM of **1a**·HCl were titrated against 0.1 M NaOH. The pH at which 0.5 eq. of NaOH was added determined the pKa (8.63; Figure S23). For the mechanistic investigations, the pH value was maintained at 8.0 by adding base as soon as the pH value dropped below this pH. The volumes pumped were used to calculate the equivalents of NaOH added in comparison to the amount of substrate added. The reaction was conducted on a 2 mL scale using 5 mM of **1a** (50 mM stock), 3.3 μM of NfsA (645 μM stock), 1.6 μM BmGDH (640 μM stock), 0.05 mM of NADP⁺ (10 mM stock) and 200 mM NaPi_i buffer (pH 8.0) with 300 mM NaCl. The reaction was stirred at 400 rpm and 25°C. After 1 hour a sample of 100 μL was taken, extracted with EtOAc and analyzed by GC-FID. The conversion measured was 73 %.

To calculate the amount of NaOH theoretically contributed by acidification based on gluconic acid production ($[H^+]_{gluconic\ acid}$) from the cofactor regeneration system when only one reduction is required, the following formula was used:

$$[H^+]_{gluconic\ acid} = [6a'] = 3.65\ mM$$

The amount of protons released by the reaction of **1a**/**1a**·HCl to **6a'** ($[H^+]_{1a \rightarrow 6a'}$) was defined as:

$$[H^+]_{1a \rightarrow 6a'} = [1a \cdot HCl]_{t_0} - [1a \cdot HCl]_{t_1} - [6a' \cdot HCl]_{t_1}$$

However, the $[6a' \cdot HCl]_{t_1}$ at pH 8.0 is neglectable because the pK_a of the protonated indazole is 1.22⁴.

$$[H^+]_{1a \rightarrow 6a'} = [1a \cdot HCl]_{t_0} - [1a \cdot HCl]_{t_1}$$

The Henderson–Hasselbalch equation was used to calculate the initial and final concentrations of protonated **1a**:

$$pH = pK_a + \log \left(\frac{[1a]}{[1a \cdot HCl]} \right)$$

With a pK_a of 8.63 for **1a** and at pH 8.0 the equation yields the following term:

$$8.00 = 8.63 + \log \left(\frac{[1a]}{[1a \cdot HCl]} \right) \Leftrightarrow -0.63 = \log \left(\frac{[1a]}{[1a \cdot HCl]} \right) \Leftrightarrow \frac{[1a]}{[1a \cdot HCl]} = 0.234$$

With the initial substrate concentration of 5 mM = $[1a]_{t_0} + [1a \cdot HCl]_{t_0}$:

$$[1a \cdot HCl]_{t_0} = 5 \text{ mM} - 0.234 \times [1a \cdot HCl]_{t_0} = \frac{5 \text{ mM}}{1.234} = 4.05 \text{ mM}$$

And after the reaction (t_1), only 27 % of substrate was left:

$$[1a \cdot HCl]_{t_1} = 5 \text{ mM} \times 0.27 - 0.234 \times [1a \cdot HCl]_{t_1} = \frac{1.35 \text{ mM}}{1.234} =$$

1.09 mM

The released acid from the transformation of **1a** to **6a** with 73 % conversion at pH 8.0 and an initial substrate concentration of 5 mM:

$$[H^+]_{1a \rightarrow 6a'} = [1a \cdot HCl]_{t_0} - [1a \cdot HCl]_{t_1} = 4.05 \text{ mM} - 1.09 \text{ mM} = 2.96 \text{ mM}$$

Considering the protons formed due to the gluconic acid production ($[H^+]_{\text{gluconic acid}}$) and the formation of **6a** from predominantly protonated **1a** ($[H^+]_{1a \rightarrow 6a'}$), we would expect 6.61 mM (1.32 eq. relative to the substrate concentration) of NaOH to be required to maintain a pH of 8.0. This is close to the 8.81 mM of NaOH (1.62 equivalents) that were actually required to maintain the pH value (Figure S24). The excess of 0.3 equivalents (2.2 mM) of NaOH can be explained by quenching of the reduced flavin cofactor by O_2 (forming H_2O_2). This reaction was observed in the absence of substrates **1a** (Figure S24). To maintain stable pH in this reaction, 3.10 mM of NaOH was required. The presented data clearly shows that indazole formation requires only one turnover of the cofactor

regeneration system. This means that the nitroso intermediate **2a** is the cyclizing intermediate, as proposed.

2. Less than 10 examples were demonstrated with only substitutions on the nitrogen or adjacent carbon. The diversity of the substrate scope is not high. For six membered rings, the scope seems to be too narrow. The extension of the nitroreductase-triggered cyclization to six-membered ring systems (e.g., cinnolines) is conceptually promising but remains underdeveloped. Please optimize the oxidation conditions (e.g., addition of a mild oxidant or use of an enzymatic oxidation system) to achieve full aromatization to cinnoline and expand the substrate scope for six-membered heterocycle formation.

We thank the reviewer for this thoughtful perspective on the substrate scope. We agree that the substrate diversity could be increased. Therefore, we included a series of ring-substituted substrates. We conducted experiments with halogen- and methoxy- in the *ortho*, *meta*, and *para* positions relative to the nitro group (substrates **1i-o**). While the enzymes converted most of the substrates very well, *ortho*-substitution seemed to lead to reduced activity due to steric hindrance. BaNTR1 appears to tolerate *ortho*-substitutions better than NfsA. Substrate **1k** was depleted by the enzymes; however, cyclization did not occur efficiently, possibly due to the strong electron-donating effect of the *para*-methoxy group, which reduces the electrophilicity of the nitroso group.

We agree that the extension of the NR-triggered cyclization to six-membered systems such as cinnolines is conceptually attractive. However, the primary focus of this study, as also indicated by the title, was the development of a robust and broadly applicable enzymatic strategy for indazole formation. Therefore, the extension to six-member rings is only included as a proof of concept and to showcase potential future developments in that direction. Within this framework, we believe that the presented substrate scope, covering diverse substitutions on the nitrogen, adjacent carbon and the aromatic ring is sufficiently representative to demonstrate the utility of the NR-triggered indazole formation. The work on enzymatic cinnoline formation is part of ongoing investigations, and further developing this route would be out of scope of this manuscript.

For substrates substituted at the aromatic ring (**1i-o**), steric hindrance seems to play an important role. Although NfsA and BaNTR1 accept the 4-chloro- and 4-methoxy-substituted substrates **1l** and **1m**, both enzymes are less active on the 3-chloro- and 3-methoxy-substituted substrates **1n** and **1o**. However, contrary to the general trend, BaNTR1 exhibits higher activity than NfsA with **1n** and **1o**. Interestingly, the 5-bromo- and 5-chloro substituted substrates (**1i** and **1j**) are well converted to the corresponding 1H-indazoles. However, when **1k** was used as the substrate, only small amounts of **6k'** were produced and the substrate disappeared overnight. At the same time, the reaction mixture and its extract turned violet; however, no new products were observed on the

gas chromatography (GC) chromatogram. This indicates that reactive nitrogen species may have formed oligomers, which is also supported by the observation of precipitate formation over time. This indicates that cyclization is hindered by the strong electron-donating effect of the para-methoxy substitution, which reduces the electrophilicity of the nitroso group.

3. Yields are moderate for some cases (6b, 6c, 6e, 6f), and high for 6d, even the conversions were the same. Could further engineering of the enzyme expand the scope of the reactions to include more diverse substrates? Though the reaction seems quite efficient now, the authors should at least attempt enzyme engineering.

We thank the reviewer for this thoughtful comment. We agree that the isolated yields vary despite comparable conversions. Although we cannot provide a definitive explanation, we note that **6d** was the only substrate that was not used as a hydrohalide salt and was therefore employed as a neat liquid. In contrast, the hydrohalide salts used for the other substrates may have contained small amounts of moisture, which could contribute to lower isolated yields despite full conversion. Additionally, the purifications were performed on a small scale, where material losses during workup and chromatography (e.g., transfer losses, adsorption on silica, extraction variability) can have a disproportionate impact on the final yield. Furthermore, we recognized while addressing another reviewer's comments that the weight of product **6d** (which was measured to be 16 mg) was not corrected by the remaining solvent as the solvent peaks were comparably low. The corrected yield was found to be 90 % (15 mg) instead of 96 %.

Regarding enzyme engineering, we agree that it could broaden the substrate scope, e.g., toward **1o**. However, given that this substrate was the only one tested that was less well accepted by the NRs, we believe that the substrate scope is sufficiently broad also using the native enzymes. Moreover, given the high conversions achieved for nearly all substrates, together with the now significantly increased substrate scope which showcases that the chosen NRs are already very efficient biocatalysts, and the focus of this study on demonstrating the feasibility of the biocatalytic concept, we believe that systematic engineering efforts would exceed the scope of this study.

4. The two enzymatic steps were performed sequentially, not simultaneously, which should be emphasized.

We thank the reviewer for this helpful remark. In the revised manuscript, we have clarified that the two enzymatic steps were performed sequentially by adding "sequential" where it was missing. For example, we added it to the description of the cascade design, the caption of Figure 6, and the conclusion.

5. Please cite the recent multienzyme stereodivergent cascade for tetrahydroquinoline synthesis (Angew. Chem. Int. Ed. 2024, 63, e202411561) to better contextualize your cascade strategy within the current literature.

We thank the reviewer for pointing out the recent relevant work. We have now cited the aforementioned paper, in the description of the cascade design section (see below). This citation provides a better context for our cascade strategy within the current literature on enzymatic cascade reactions for heterocyclic pharmaceutical building blocks.

Furthermore, one-pot enzymatic cascades including IREDs and NRs have recently been shown to be highly efficient for synthesizing heterocyclic pharmaceutical building blocks⁴².

6. Three replicates were mentioned, but no error bars or statistical analyses are provided. Please add mean \pm SD (or SEM) to all bar/line charts and clearly state the sample size (n) and statistical methods used in the figure captions.

We thank the reviewer for this important point. Mean \pm SD and the corresponding sample size have been added to the figures in which replicate experiments were performed (e.g. Figure 2 and S12-16). Statistical annotations were included where applicable.

Example:

Figure S14. Kinetic studies of NfsA towards NADPH in the presence of 1 mM **1a**. GC areas of **6a'** are shown. *Mean \pm SD; n = 3. Non-linear regression: Michaelis-Menten function with an orthogonal distance regression using OriginPro2020 (OriginLab Corporation, Northampton, MA, USA).*

7. Please report the total reaction volume (in x mL) for each experiment in the main text.

We have added the total reaction volume for each experiment to the figure captions and the method sections to ensure full clarity and reproducibility.

8. Supply ¹³C NMR data for all new compounds (**6f**). Ensure consistent formatting for NMR solvent descriptions throughout the Supporting Information (e.g., “400 MHz, CDCl₃”).

We thank the reviewer for this comment. We have added the missing ¹³C NMR data for compound **6f** (and all other compounds isolated) and ensured consistent formatting of all NMR solvent descriptions throughout the Supporting Information.

We are confident that these changes made in response to the reviewer comments and suggestions have strengthened the manuscript considerably. Once again we thank you for your constructive comments and hope that you are able to accept our manuscript for publication in *Nature Communications* in light of the above points.

Sincerely,

Sandy Schmidt

(on behalf of all the authors)

Response to reviewers' comments

We are grateful for the enthusiastic feedback from the reviewers and are delighted that all three recommend our manuscript for publication. We have uploaded a revised version of the paper that addresses reviewer 1's final (minor) comments.

The specific changes made are outlined in the reviewer response file below (author responses are in blue) and highlighted in yellow in the revised manuscript. We used the track changes function of Word for changes made to the manuscript based on editorial requests.

Reviewer #1 (Remarks to the Author):

In their revised manuscript, Schmidt and co-workers have thoroughly addressed the minor concerns voiced in my original reviewer report, as well as the criticism raised by the other two reviewers. The authors have carried out several additional experiments, the most significant of which are an extension of the substrate scope to ring-substituted derivatives and a titration experiment that supports their original mechanistic proposal. In addition, many small improvements were implemented, ranging from determination of GC response factors to an increased level of detail in figure captions and experimental procedures, and the correction of typos and minor terminological issues. All these revisions have further strengthened the manuscript, and I recommend its publication in Nature Communications after a last round of minor corrections, which can be implemented without further peer review. Suggestions for these final improvements are provided below:

(1) Terminology: Please ensure consistent spelling of the term "side product" (with a space, not a hyphen).

We thank the reviewer for these helpful comments. The spelling of the term "side product" has been made consistent throughout the manuscript.

(2) Listings of GC-MS data: It is not common practice to report expected and observed masses in unit-resolution GC-MS analysis. Instead, provide a list of all significant mass fragments including their relative abundance, with identification of the parent ion (M^+); e.g., for 6a': GC-MS (EI, 70 eV): $m/z = 118 (M^+, 100)$, 91 (31), 63 (40), 52 (19).

The GC-MS data listings were revised to report significant mass fragments with their relative abundances and identification of the parent ion (M^+).

(3) GC-MS analysis conditions: The flow rate of 13.8 mL/min appears unusually high. I assume it is the total flow (including split flow, and septum purge flow, if applicable) that is reported here, but in order to allow other researchers to reproduce the GC separations, the column flow rate (the part of total carrier gas flow that is directed through the column) should be reported.

The GC–MS analysis conditions were corrected by replacing the total flow rate with the column flow rate (1.81 mL/min), which has now been reported to ensure reproducibility of the GC separations.